# An approach to identify the best climate models for the assessment of climate change impacts on meteorological and hydrological droughts

Antonio-Juan Collados-Lara [1], Juan-de-Dios Gómez-Gómez [2], David Pulido-Velazquez [1], Eulogio Pardo-Igúzquiza [3]

[1]Spanish Geological Survey (IGME), Urb. Alcázar del Genil, 4. Edificio Zulema, Bajo, 18006 Granada, Spain
[2]Spanish Geological Survey (IGME), La Calera, 1, 28760 Tres Cantos, Spain
[3]Spanish Geological Survey (IGME), Ríos Rosas, 23, 28003 Madrid, Spain

*Correspondence to*: Antonio-Juan Collados-Lara (ajcollados@gmail.com)

**Abstract.** This paper describes the benefits of using more reliable local climate scenarios to analyse hydrological responses. It assumes that Regional Climate Model (RCM) simulations are more reliable when they provide better approximations to the historical basic and drought statistics after applying a bias correction to them. We have investigated whether the best solutions in terms of their approximation to the local meteorology may also provide the best hydrological assessments. We have carried out a classification of the corrected RCM simulations used for both approximations. This has been applied in the Cenajo Basin (south-eastern Spain), where we show that the best approximations of the historical meteorological statistics also provide the best approximations for the hydrological ones. The selected RCMs were used to generate future (2071-2100) local scenarios under the RCP 8.5 emission scenario. The two selected RCMs predict significant changes in mean precipitation (-31.6 % and -44.0 %) and mean temperature (+26.0 % and +32.2 %). They also predict higher frequency (from 5 events in the historical period to 20 and 22 in the future), length (4.8 to 7.4 and 10.5 months), magnitude (2.53 to 6.56 and 9.62 SPI) and intensity (0.48 to 1.00 and 0.94 SPI) of extreme meteorological droughts. These two RCMs also predict higher changes in mean streamflow (-43.5 and -57.2 %) and hydrological droughts. The two RCMs also predict worrying changes in streamflow (-43.5 % and -57.2 %) and hydrologically extreme droughts: frequency (from 3 to 11 for the first and 8 events for the second model), length (8.3 to 15.4 and 29.6 months), magnitude (from 3.98 to 11.84 and 31.72 SSI), and intensity (0.63 to 0.90 and 1.52 SSI).

## 1 Introduction

During the last decades large scale intensive droughts have been observed in all the continents around the globe (Kogan and Guo, 2016). In Europe the 2003 and 2015 droughts may be regarded as the most extreme droughts over the last 250 years (Hanel et al., 2018). In Spain the 2005 drought was the most marked since records began (Vicente-Serrano et al., 2017).

Since 1950 several indices have been proposed in the literature to assess different types of droughts by studying different climatic and hydrological variables (Heim, 2002; Mishra and Sight, 2010; Pedro-Monzonis et al., 2015). For instance, we have the Palmer Drought Severity Index (PDSI) (Palmer, 1965), the Crop Moisture Index (CMI) (Palmer, 1968), the

Standardized Precipitation Index (SPI) (McKee et al., 1993), the Soil Moisture Drought Index (SMDI) (Hollinger et al., 1993), the Vegetation Condition Index (VCI) (Liu and Kogan, 1996) and the Standardized Precipitation Evapotranspiration Index (SPEI) (Vicente-Serrano et al., 2010). From their names we can deduce that some of them were defined to analyse specific characteristics, such as length, magnitude and intensity, and different types of droughts (meteorological, agricultural or hydrological droughts). Some of these indices can be generalized to analyse most of the characteristics of the different types of droughts, (as for example the SPI, Mckee et al., 1993, 1995), and their propagation.

In most of the scarce water areas droughts will become intensified in the future due to global change, which is associated with an increment in the occurrence of extreme events. The impact of global change on droughts is a major concern of climate change. The Mediterranean basin is one of the areas which will be most affected by droughts in the future (Tramblay et al., 2020). In addition, the latest climate change studies expect significant decreases in resources in the Mediterranean basins, which will cause a significant environmental, economic and social impact (Cramer et al., 2018). Although in the recent years the number of papers related to this issue has increased (Marcos-Garcia et al., 2017; Collados-Lara et al., 2018), we still need to make advances on the assessment (through appropriate indices and techniques) of this important social issue (Mishra and Sight, 2011). Some authors directly use Regional Climate Model (RCM) simulations to assess future droughts (e.g. Lloyd-Hughes et al., 2013; Zhang et al., 2017) in water resource systems, other studies show cases with significant bias between the historical and the modelled values (Cook et al., 2008; Seager et al., 2008), which requires further analysis and corrections.

Different approaches [e.g. delta change (Pulido-Velazquez et al., 2018) or bias correction (Collados-Lara et al., 2019)] can be used to downscale RCM simulations according to the local historical climatology (Collados-Lara et al., 2018). They use different statistical correction techniques [e.g. first and second moment correction, regression (Collados-Lara et al., 2020) or quantile mapping (Gudmundsson et al., 2012)]. The different approaches produce different approximations of the statistics of the historical period depending on the RCM simulations. They also show a wide range of future corrected simulations that reveals the uncertainty related with the climate models and their propagation (Pardo-Igúzquiza et al., 2019; Pulido-Velazquez et al., 2018). Hence, the use of several RCMs is recommended to assess the impact of climate change.

The generated scenarios can be used to define a set of individual local projections, which take into account the uncertainty, or to create ensembles of them, which define more robust climate scenarios than those based on a single projection (AEMET, 2009). In both cases a classification of RCM simulations according to their reliability in terms of their capacity to approximate historical meteorological statistics is needed. Depending on the objective of the study the reliability classification should consider different statistics. For drought assessment, in addition to the basic statistics (mean, standard deviation, and skew coefficient), drought statistics (e.g. frequency, duration, magnitude, and intensity) should be studied (Collados-Lara et al., 2018). In literature there are few studies which analyse the reliability of RCMs for considering meteorological droughts (Peres et al., 2020; Aryal and Zhu, 2021). In this study we also analyse the propagation of meteorological droughts to hydrological droughts. To the best of our knowledge, there are no studies that have analysed whether climate models that provide the best approximations of the local historical meteorology may also provide better

assessments of the hydrological impact. In these cases, the generated local climate scenarios have to be propagated by using hydrological models (Senent-Aparicio et al., 2018; Pardo-Igúzquiza et al., 2017).

The main objective of this paper is to answer the following question: Do climate models that allow better approximations of local meteorology improve the assessment of hydrological responses? It is a question that will be answered by a novel approach based on the analyses of basic and drought statistics. We propose a classification method for RCM simulations

according to their capacity to generate local climate scenarios that reproduce the historical period (in terms of basic and drought statistics). The classification has been done and compared for both meteorological and hydrological scenarios, considering basic and drought statistics, in order to compare the results for both types of droughts. Based on these analyses, an integrated statistical method is proposed to generate "more reliable" potential future climate scenarios from RCM simulations and historical data. Our aim is to contribute to a better assessment of future meteorological and hydrological

droughts and is applicable to any case study.

The paper is structured as follows. In section 2 we describe the proposed method. Section 3 is focused on the description of the case study and the available data, including historical and climate simulations. In section 4 the results and in section 5 the discussion. Finally, section 6 summarizes the main conclusions of this study.

## 2 Methodology

The steps that define the proposed method are represented in the flow chart shown in Figure 1. It requires compilation of the monthly information about historical precipitation, temperature and streamflow within the system, and RCM simulations. Long historical series are needed in order to perform an appropriate statistical analysis of the proposed approaches. Periods of analyses that cover 30 years or even longer are recommended. A statistical analysis is proposed in order to assess the bias between the statistics of the RCM control simulations and the historical scenarios in the case study. If there are significant

differences between them, the RCM simulations for the future horizon cannot be directly used to define the future local climate scenarios of the system, and we need to apply some statistical corrections to them.

### 2.1 Correction of historical climate scenarios

We have corrected the RCM control simulations by applying a bias correction approach. It is based on a transformation function that minimizes the differences between the statistics of the control simulations and the historical scenarios (Shrestha

et al., 2017). The statistical transformation was defined by a quantile mapping technique based on empirical quantiles. We used the open-source R package qmap (Gudmundsson et al., 2012). Quantile mapping with empirical quantiles uses a non-parametric transformation function. In this approach the empirical cumulative distribution functions (CDFs) are approximated using tables of empirical quantiles. It estimates the values of the empirical CDFs of observed and simulated time series for regularly spaced quantiles to create the table that relates the observed and simulated time series (Enayati et al.,

2021). The values between the percentiles are approximated by using linear interpolation. These interpolations are used to

adjust a datum with unavailable quantile values. We have used its table of empirical quantiles for each month of the year. These tables, (which are obtained by using the CDF of the observed and simulated values from RCMs), are also used to correct future simulations (from RCMs). If the RCM values are greater than the historical ones used to estimate the empirical CDF, the correction found for the highest quantile of the historical period is used (Gudmundsson et al., 2012). This technique has been demonstrated to be better than other simpler ones (first and second moment correction, regression, and quantile mapping using parametric distributions) to correct basic statistics (mean, standard deviation, and skew coefficient) (see Collados-Lara et al., 2018). This is why we have chosen quantile mapping (using empirical quantiles) for this study.

## 2.2 Definition of the rainfall-runoff model

A hydrological balance model is defined to propagate different climate scenarios (historical, control, corrected control and futures) in order to assess hydrological series (streamflow series) and their basic and drought statistics. A rainfall-runoff model was calibrated and validated (by minimizing the sum of the squared errors for each month) with the available historical data (Pulido-Velazquez et al., 2008). In this study we have applied a Temez model (Témez, 1977) to assess inflow scenarios in natural flow regime in the basin. It is a lumped conceptual hydrological model frequently used in Spanish basins (Escriba-Bou et al., 2017, Perez-Sanchez et al., 2019). It is formulated by balance and transfer equations using just four parameters and two storage tanks (representing the soil or unsaturated zone and the aquifer). The potential evapotranspiration, which is required for this model, has been estimated by applying the Thornthwaite method (Thornthwaite, 1948) from temperature data.

## 2.3 Classification of the RCMs

An analysis of the performance for each RCM simulation after applying the statistical correction was performed for both the meteorological series and the hydrological simulations. The accuracy of the model was analysed in terms of basic (mean, standard deviation, and skew coefficient) and drought statistics (frequency, duration, magnitude, and intensity). The meteorological and hydrological drought analysis was developed by applying the Standard Precipitation index (SPI) (Bonaccorso et al., 2003; Livada and Assimakopoulos, 2007) and Standard Streamflow index (SSI) (Salimi et al., 2021), respectively. They were estimated for periods of aggregation equal to 12 months. The calculation method requires the transformation of a gamma frequency distribution function to a normal standardized frequency distribution function. The statistics of the SPI/SSI series are obtained by applying the run theory (González and Valdés, 2006; Mishra et al., 2009) for different SPI/SSI thresholds from the lower SPI/SSI to 0. The frequency is defined as the number of droughts events for each SPI threshold. We have assessed the duration of each drought event as the number of months that the SPI is below a given threshold, its magnitude as the summation of the SPI values for each month of the event, and its intensity as the minimum SPI value. For each threshold we have estimated the mean duration, magnitude, and intensity as the mean values of the cited variables for all the drought events. The probability of the occurrence of precipitation or streamflow for the SPI/SSI calculation, in the corrected control and future simulations, was obtained by using the parameters calibrated from the

observed series, in order to perform an appropriate comparison (Marcos-Garcia et al., 2017). In order to analyse the benefit of the proposed method to select future climate scenarios in the assessment of basic and drought statistics, we checked whether the local climate scenarios from RCM simulations that allow better approximations of the meteorology did provide better assessments of the hydrological statistics.

We assessed the performance for each RCM in the reference period by applying the following error index (SE):

$$SE = \frac{1}{\left(\frac{1}{N}\sum_{i=1}^{N} S_{h,i}\right)^2} \frac{1}{N} \sum_{i=1}^{N} \left(S_{c,i} - S_{h,i}\right)^2$$

(1)

where S is the statistic being considered, N is 12 in the case of basic statistics (number of months in a year) and the number of SPI thresholds considered in the case of droughts statistics, c is corrected control scenario, and h is the historical scenario. Note that this index is a mean squared error of the corrected control with respect to the historical values. It is divided by the square of the mean historical value in order to make the results comparable for different statistics.

This error index was calculated for each basic (mean, standard deviation, skew coefficient) and drought (frequency, length, magnitude, and intensity) statistic and used to classify RCMs according to their reliability for the assessment of meteorological and hydrological impact. For each statistic we classified the RCMs according to the following criteria. The RCMs that have a SE lower than 0.0009 (equivalent to a 3 % of relative error) are not penalized. The rest of RCMs are penalized proportionally from 1 to 10 with 1 for being the lowest SE and 10 for the highest SE. Finally, we carried out the classification of the RCMs for the meteorological and hydrological analyses by calculating the average of the penalization for all the statistics (basic and drought). The penalization approach allows us to define an index (SE) threshold below which the RCMs are not penalized. It also allows us to give similar weight to all the statistics in the final classification. Note that the skew coefficient and drought statistics have higher SE values. If we add up the SE values for all the statistics and we classify RCMs in accordance with this total, the mean or standard deviation statistics will not influence in the final classification.

## 2.4 Generation of local future climate scenarios and statistical analysis of the results

The classification of local climate scenarios from RCM simulations allows us to identify approaches with higher reliability for both meteorological and hydrological statistics. For these RCM simulations we can generate local future climate scenarios by applying the same transformation function used to correct the control simulation to the future simulation series. These scenarios can be used as individual projections that allow us to take into account the uncertainty by considering a set of different RCM simulations. An ensemble of scenarios could be also applied to produce more robust climate scenarios than those based on a single projection. Finally, these future scenarios were analysed in terms of basic and drought statistics, and compared with the historical scenarios to assess the impact of climate change on meteorology and hydrology.

**3 Case study and data**

The proposed methodology was applied to the Cenajo basin. It is located in south-eastern Spain (Fig. 2), within the basin headwaters of the Segura River, which is the main stream of the Segura basin. The main cities of the system are Murcia with a population of around 440,000, and Alicante with a population of more than 330,000. These cities are partially supplied by the Segura River system. The Segura River is also important for agriculture. The main socioeconomic activity is the irrigated agriculture, traditionally concentrated in the alluvial and coastal plains. The main crops are citrus and fruit trees, and also green and other vegetables. This coastal basin is an example of a Mediterranean area with a significant water demand, mainly for irrigation but also for urban supply (with an important seasonal component for the tourist sector), and low availability of resources. In fact, it is a system with significant deficits that needs water transfers from Tagus Basin and additional supplies from desalination plants to meet the existing demands. The Cenajo basin has a Mediterranean climate. In the period 1972-2001, the mean annual precipitation was 623.6 mm and the mean temperature 14.0 °C. In the same period the mean annual streamflow was 443.6 Mm³. This is a critical area where climate change will exacerbate these problems by reducing the availability of resources and increasing irrigation requirements. It will also cause an increase in the magnitude and frequency of extreme events, such as droughts.

We used historical climate data (precipitation and temperature) provided by the Spain02 v2 dataset (Herrera et al., 2012) for the period 1972-2001. In this study we have carried out a lumped analysis in the Cenajo basin. The RCMs were retrieved from the CORDEX project (CORDEX PROJECT, 2013), with a spatial resolution of 0.11° (approximately 12.5 km). Note that Spain02 dataset uses the same reference grids as the CORDEX project. The most pessimistic emission scenario (RCP8.5) for the future horizon 2071-2100 was selected for the future projections. For this scenario we analysed nine RCMs corresponding to four different General Circulation Models (GCMs) (see Table1). In our case study 33 cells of the grid mesh fall within the basin. The historical and simulated (from RCMs) precipitation and temperature were aggregated at basin scale considering a weighted average value according to the area of each grid mesh inside the basin. We also used official monthly natural streamflow data within the Cenajo basin for the historical period 1972-2001 (adopted as reference). The SIMPA model streamflow series (Alvarez et al., 2005) were used as historical data for calibration, due to the highly altered flow regime measured in gauge stations within this basin. Note that in the studied basin there are several dams. SIMPA is the model used by the water authorities in Spain for water planning. It was calibrated previously by restoring the gauge stations to the natural regime. Therefore, we assessed inflow scenarios in natural flow regime in the basin. These data were taken from the available information from the Spanish Ministry for Agrarian Development and Irrigation.

# 4 Results

## 4.1 Rainfall-runoff model

The rainfall-runoff model for the Cenajo basin was calibrated and validated using the available monthly climate data (precipitation, temperature, and potential evapotranspiration) and streamflow data for the period October 1971 to September 2007. We divided the series with available data in two periods to perform a calibration (from October 1971 to September 1989) and validation (October 1989 to September 2007) of the model. The performance of the model was assessed by using the Nash-Sutcliffe efficiency (NSE) coefficient, the correlation coefficient ($R^2$), and the root mean squared error (RMSE). These statistics and the historical and simulated streamflow series are shown in Figure 3a. For the entire period (October 1971 to September 2007) the performance is also good (NSE = 0.94) and it is higher (NSE = 0.96) if we focus on the monthly mean within the mean year for the entire period (Fig. 3b). The model was used to propagate the impact of climate variables on the streamflow between 2071 and 2100, a 30-year horizon, which is a period of time usually used in climate change analysis.

## 4.2 Corrected historical simulations

The observed differences between the historical series and the control simulation series of precipitation and temperature for the reference period (1972-2001) in terms of basic statistics are significant (see Table 2). The relative differences between the historical and the control simulations for the mean yearly precipitation (Fig. 4a) vary from -5.6 % for RCM5 and 52.8 % for RCM8. In the same way, the distances in the standard deviation (Fig. 4c) and skew coefficient (Fig. 4e) are also great. The relative differences between the historical temperature and the control simulations for the mean year values (Fig. 5a) vary from -6.2 % for RCM3 and -39.4 % for RCM5. The differences in the temperature standard deviation (Fig. 5c) and skew coefficient (Fig. 5e) are also remarkable. These differences force us to apply the correction approach defined in section 2.1 for all the RCMs considered. It uses the CDF (quantiles) of the historical series and the control series obtained from the RCM simulations to perform the correction. The precipitation and temperature quantiles of the observed and control simulation series of RCM1 in the reference period are shown in Figure 6. The same information was generated for all the RCM simulations and used to correct the RCM outputs. The corrected control simulation series presents a very good fit with respect to the historical series in terms of basic statistics for precipitation [mean (Fig. 4b), standard deviation (Fig. 4d) and skew coefficient (Fig. 4f)] and for temperature [mean (Fig. 5b), standard deviation (Fig. 5d) and skew coefficient (Fig. 5f)]. The differences between the historical series and the corrected control simulation for the basic statistics are close to zero. The differences in mean annual values are negligible (see Table 2). This confirms the results obtained by Collados-Lara et al. (2018) when they compared different statistical correction techniques. The quantile mapping (with empirical quantiles) technique shows very good results in terms of the basic statistics when the RCMs are corrected.

The same analysis of basic statistics was done for the streamflow (Fig. 7). The relative differences between the historical and the control simulations for the mean yearly streamflow (Fig. 7a) vary from -4.9 % for RCM5 and 125.5 % for RCM8. It also

shows very large differences for the standard deviation (Fig. 7c) and the skew coefficient (Fig. 7e). The fit of the corrected control simulation series of streamflow to the historical series is not as good as for precipitation and temperature, but a remarkable improvement is observed. The reason could be that we are neglecting the inter-variable dependence of climate variables and not taking into account the dependence between precipitation and temperature when the bias correction is applied. Therefore, some differences might appear in the streamflow that depend on the combined interaction of both variables. The relative differences for the mean streamflow in the case of the corrected control simulation (Fig. 7b) vary from -1.8 % for RCM2 and -4.6 % for RCM8. Similar improvements are observed for standard deviation (Fig. 7d) and skew coefficient (Fig. 7f).

In the case of the meteorological droughts (calculated from SPI) the bias correction approach clearly improves the fit of the RCM simulation series to the historical series for the four considered statistics (frequency, duration, magnitude and intensity). Note the differences between the left-hand panel of Figure 8 (control simulation and historical series) and the right-hand panel of Figure 8 (corrected control simulation and historical series). For frequency the mean of SE for all the RCMs before the correction is 0.69 and after the correction is 0.23. For duration, magnitude and intensity these values are respectively 0.51 vs. 0.17, 0.88 vs. 0.30 and 0.38 vs. 0.13. In the same way, hydrological droughts were studied considering the SSI.. Significant improvements are also observed for hydrological droughts (Fig. 9) after the bias correction procedure: frequency (mean SE of 0.63 vs. 0.34), duration (mean SE of 0.50 vs. 0.23), magnitude (mean SE of 0.83 vs. 0.51), and intensity (mean SE of 0.48 vs. 0.15). The left-hand panel represents the drought statistics of the historical and control series before applying the bias correction technique and the right-hand panel after a bias correction approach.

**4.3 Classification of RCMs**

The classification of RCMs (after the bias correction of the simulations) is based on the approximation of the meteorological and hydrological statistics (basic and drought statistics) by applying the procedure described in section 2.3 and is included in Table 3. The two best-corrected RCMs for meteorology (RCM2 and RCM9) are also the best models for hydrological assessment (maintaining the first and second position in both cases). Nevertheless, the third "best" model for meteorology is the fifth in hydrological assessment, and the fourth in meteorology and the third in the hydrological assessment. Although they are still in the group of the best approaches, it demonstrates that there is not a cause-effect relationship; a better meteorological approximation does not always mean a better hydrological assessments. We have only demonstrated that, in our case study, the RCMs that provide the best approximations of the meteorology also provide the best assessments of the hydrological impact.

**4.4 Corrected future local scenarios**

The corrected RCM2 and RCM9, which are the best climate models to reproduce historical meteorology and hydrology, were used to generate local potential scenarios of precipitation and temperature. The rainfall-runoff model was used to propagate the impact of climate variables on streamflow. In order to compare the historical and the future scenarios basic and

drought statistics were analysed for the horizon 2071-2100. The considered RCMs predict significant reductions of mean precipitation (-31.6 % and -44.0 % for RCM2 and RCM9 respectively) and an increase in mean temperature (26.0 % and 32.2 % for RCM2 and RCM9 respectively) (see Fig. 10a and 10b respectively). The average change in monthly standard deviation of precipitation is -6.2 % and -32.3 % for RCM2 and RCM9 respectively. In the case of temperature these changes are 23.9 % and 4.8 %. Both RCMs predict a decrease in the standard deviation in precipitation and an increase in the standard deviation of temperature in the future (see Figs. 10c and 10d respectively). However the expected values of the changes are significantly different. Both RCMs also predict significantly different changes in the skew coefficient of series (Fig. 10e and 10f). With respect to the hydrology analysis, both RCMs predict significant decreases in mean streamflow (-43.5 % and -57.2 % for RCM2 and RCM9 respectively) (Fig. 11a). In the case of the standard deviation, the RCMs predict a reduction (Fig. 11b). The average change in monthly standard deviation is -26.2 % and -57.5 % for RCM2 and RCM9 respectively. In the case of the skew coefficient both RCMs show an increment with respect to the historical scenario (Fig. 11c). We also analysed the coefficient of variation (ratio of the standard deviation to the mean) of historical and future series of precipitation, temperature, and streamflow (Table 4). Both RCMs predict an increase in the precipitation and streamflow variability, and a reduction in temperature variability. This increment in precipitation variability is also described in other climate change impact studies (Pendergrass et al., 2017; Polade et al., 2017).

Significant changes are also expected for droughts. In the case of the meteorological droughts the first SPI threshold for which drought periods are detected in the historical scenario is -3.0. In the future scenarios this value is -5.2 and -4.6 for RCM2 and RCM9 respectively (Fig. 12). In order to perform an appropriate analysis of the future droughts with respect to the historical, the future SPI calculation was estimated by using the parameters of the gamma distribution obtained in the historical period (Collados-Lara et al., 2018). If the parameters of the gamma distribution were adjusted to the future series of values, the changes in the parameters would be significant. For RCM2 we would obtain $\alpha = 19.9$ and $\beta = 2.6$ (instead of the historical values $\alpha = 16.1$ and $\beta = 3.2$) and for RCM9 $\alpha = 19.0$ and $\beta = 2.7$ (instead of the historical values $\alpha = 16.1$ and $\beta = 3.2$). The maximum frequency of meteorological droughts in the historical period is obtained for the SPI threshold of 0 while in the case of the future scenarios this is obtained for -1.1 and -1.9 for the RCM2 and RCM9 respectively. For the threshold of -1.7 of SPI (considered to define extreme droughts in the Droughts Plan of the Segura River basin authority) in the historical period 5 drought events are detected with a mean length of 4.8 months. The mean magnitude and intensity of these events are 2.53 and 0.48 SPI. In the case of the future scenario of the RCM2 20 drought events are detected with a mean length, magnitude and intensity of 7.4 months, 6.56 and 1.00 SPI. The case of the future scenario of RCM9 is even more worrying with 22 extreme drought events which have a mean length, magnitude and intensity of 10.5 months, 9.62 and 0.94 SPI respectively.

In the case of the hydrological droughts the first SSI threshold in which we detected droughts is -2.9 (similar to the meteorological droughts). In the future scenarios this value is -3.9 and -4.2 for the RCM2 and RCM9 respectively (Fig. 13). For the threshold -1.7 of SSI significant changes are expected for both RCMs with respect to the historical period: frequency (from 3 to 11 and 8 events), length (8.3 to 15.4 and 29.6 months), magnitude (from 3.98 to 11.84 and 31.72 SSI), and

intensity (0.63 to 0.90 and 1.52 SSI). Note that in the case of the hydrological droughts the minimum SSI in the future scenario is obtained for the RCM9 and in the case of the meteorological droughts the minimum SPI is obtained for the RCM2. However in both cases (meteorological and hydrological) the RCM9 shows a higher impact on the mean length, magnitude and intensity of the drought events.

## 5 Discussion

The selected RCM simulations cannot be used directly for the case studied due to the detected biases. The relative differences vary in the range -5.6 % to 52.8 % for precipitation and -6.2 % to -39.4 % for temperature. It is accepted in the scientific community that RCMs must be corrected to adapt them to the local climate conditions (Teutschbein and Seibert, 2012).

In this study we have used the quantile mapping based on the empirical quantile technique to perform the bias correction of the RCMs. A previous comparative analysis of different correction techniques (first moment correction, first and second moment correction, regression, and quantile mapping using distribution and empirical quantiles) demonstrated the higher accuracy of the empirical quantile mapping (Collados-Lara et al., 2018). This technique is able to provide very good results to correct basic statistics (mean, standard deviation and skew coefficient) as we have confirmed in this study. However, some authors argue that using simple techniques as linear scaling is sufficient for hydrological analysis at a monthly resolution (Shrestha et al., 2017). Other authors assumed that a first and second moment correction is sufficient for hydrological applications (Collados-Lara et al., 2019). This topic is still open to discussion in the scientific community and authors are even developing and testing new techniques [e.g. TIN-Copula (Lazoglou et al., 2020), Markov chains (Liu et al., 2020)].

Another aspect brought up in this paper in the generation of local future scenarios is the selection of the RCMs. In this study we have proposed a method to classify the RCM simulations based on basic and drought statistics of the corrected series. Collados-Lara et al. (2018) proposed a multi-criteria analysis to rule out the worse approximations. In this paper our aim has been to classify all the corrected RCM simulations according to their capacity to reproduce the historical statistics. On the other hand, the proposed method also considers hydrological statistics, also including droughts. We have shown in a case study that the corrected RCM simulations that provide the best approximations of the meteorological statistics also provide the best approximations for the hydrology.

Finally, we have also shown that the best corrected RCMs for reproducing the climate and hydrological conditions in the reference period may provide significant differences in the assessment of the impact of future climate change, due to the high uncertainty related with the RCM simulations of future potential scenarios (Sørland et al., 2018). Depending on the case study, the proposed analyses and classification (based on the reference period) can be used to identify the more reliable individual projections for the future period. It will allow us to define sets of selected individual projections to take into account future impact uncertainty by considering the most "reliable" corrected RCMs (Pardo-Igúzquiza et al., 2019). It also

allows us to define an ensemble of scenarios defined by the selected corrected RCM simulations, which could produce more robust climate scenarios than those based on single projections (Fowler et al., 2007).

## 5.1 Hypotheses assumed, limitations and future research

Although we have demonstrated the utility of the proposed approach to assess the future impact on meteorological and hydrological droughts, we want to highlight some hypotheses and limitations assumed and to identify potential future
research aligned with this study:

- We have used a bias correction method based on the assumption of bias stationarity of climate model outputs. However, this assumption may not be valid for studying some problems due to the significance of the influence of climate variability on them. Other approaches should be explored to take into account the non-stationarity bias of RCM simulations (e.g. Hui et al., 2020).
- We have applied the same bias correction procedure for all the range values in accordance with the climate variable distribution function. We have not considered the impact of bias correction techniques on the tails of the distribution, which could be important to analyse extremes (Volosciuk et al., 2017).
- In this study a univariate bias correction method is used. It does not consider the dependence between precipitation and temperature which could be explored in future assessments. Meyer et al. (2019) found that incorporating or
ignoring inter-variable relationships between temperature and precipitation could impact the conclusions drawn in hydrological climate change impact studies in alpine catchments.
- The streamflow information available for this case study cannot be divided into two long-enough (e.g. 30 years) series representative of the climate/hydrology to perform explicitly a validation of the bias correction models (Chen et al., 2021). We have assumed that the statistics of any long-enough periods remain invariant. In this case the
calibration implicitly could be considered validated, due to the fact that the same results would be obtained under this hypothesis for any other period representative of the climate/hydrology conditions.
- In our case study the influence of temperature was considered only in the hydrological assessment by using rainfall-runoff models. However other meteorological drought indices that consider temperature could be included in the analysis [e.g. the Standardised Precipitation-Evapotranspiration Index (SPEI) (García-Valdecasas Ojeda et al.,
2021)].
- The corrected control simulation series obtained by using a quantile mapping bias correction presents a very good performance with respect to the historical series in terms of basic statistics. In the case of droughts (calculated from SPI/SSI) the bias correction approach clearly improves the fit of the RCM simulation series to the historical series, but the performance is lower than for the basic statistics. Other bias correction procedures should be explored to
improve the performance of drought statistics.
- The proposed method has not been tested in other typologies of basin, such as for example in Alpine basins where the snow melt component may have a significant influence on the results.

## 6 Conclusions

In this study we have proposed a method to classify the corrected RCM simulations according to their capacity to reproduce the historical statistic. It considers basic (mean, standard deviation, and skew coefficient) and drought statistics (frequency, length, magnitude, and intensity) of the meteorological and hydrological series, and could be applied to any case study. We have also shown that the corrected RCM simulations that provide the best approximations of the meteorology also provide the best assessments of the hydrological impact.

The two best classified corrected RCM simulations were used to generate potential local scenarios of precipitation, temperature and streamflow by using a lumped hydrological model. These projections were used to assess the impact of climate change on local meteorology and hydrology within the Cenajo basin (south-eastern Spain). We analysed the change in basic and drought statistics. The selection of corrected RCM simulations predict a significant future impact on mean precipitation (-31.6 % and -44.0 %) and an increase in mean temperature (26.0 % and 32.2 %). They also predict a higher

frequency (from 5 to 20 and 22 events of droughts), length (from 4.8 to 7.4 and 10.5 months), magnitude (from 2.53 to 6.56 and 9.62 SPI) and intensity (from 0.48 to 1.00 and 0.94 SPI) of extreme meteorological droughts. These changes are also propagated to the hydrological droughts. The studied area is located in the headwaters of the Segura River where the basin is an example of a Mediterranean area with a significant water demand, mainly for irrigation but also for urban supply, and low availability or resources. In these places the methodologies to assess the impact of climate change on droughts are useful

tools for water resource policy and decision makers.

**Code/data availability**

The historical climate data are available in the web site of the Spain02 project, the RCMs are available in the web site of the CORDEX project, and The SIMPA model streamflow series were taken from the available information from the Spanish Ministry for Agrarian Development and Irrigation.

The codes used in this study are available from the corresponding author upon reasonable request.

**Author contributions**

AJCL, JDGG, and DPV designed and conducted the research under the supervision of EPI, AJCL and JDGG performed the calculations and prepared the figures under the supervision of DPV and EPI, all authors wrote the paper and prepared the revisions. DPV obtained the funding. The final version has been approved by all co-authors.

**Competing interests**

The authors declare that they have no conflict of interest.

**Funding**

This research was partially supported by the research projects SIGLO-AN (RTI2018-101397-B-I00) from the Spanish Ministry of Science, Innovation and Universities (Programa Estatal de I+D+I orientada a los Retos de la Sociedad) and
GeoE.171.008-TACTIC from GeoERA organisation funded by European Union's Horizon 2020 research and innovation programme.

**Acknowledgments**

We would like to thank the Spain02 and CORDEX projects for the data provided and the open-source R package qmap.

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

**Table 1: Regional and global climate models considered.**

|        | RCM        | Nested to GCM  |
|--------|------------|----------------|
| RCM1   | CCLM4-8-17 | CNRM-CM5       |
| RCM2   | CCLM4-8-17 | EC-EARTH       |
| RCM3   | CCLM4-8-17 | MPI-ESM-LR     |
| RCM4   | HIRHAM5    | EC-EARTH       |
| RCM5   | RACMO22E   | EC-EARTH       |
| RCM6   | RCA4       | CNRM-CM5       |
| RCM7   | RCA4       | EC-EARTH       |
| RCM8   | RCA4       | MPI-ESM-LR     |
| RCM9   | WRF331F    | IPSL-CM5A-MR   |

**Table 2: Mean annual values of precipitation and temperature for the historical and the RCM simulations (and corrected RCM simulations) in the reference period (1972-2001).**

|            | Mean annual precipitation (mm) | Mean annual corrected precipitation (mm) | Mean annual temperature (ºC) | Mean annual corrected temperature (ºC) |
|------------|--------------------------------|------------------------------------------|------------------------------|----------------------------------------|
| Historical | 623.6                          | -                                        | 14.0                         | -                                      |
| RCM1       | 700.5                          | 623.5                                    | 10.4                         | 14.0                                   |
| RCM2       | 550.7                          | 623.1                                    | 10.4                         | 14.0                                   |
| RCM3       | 503.6                          | 623.3                                    | 13.2                         | 14.0                                   |
| RCM4       | 571.7                          | 623.6                                    | 10.1                         | 14.0                                   |
| RCM5       | 588.7                          | 623.3                                    | 8.5                          | 14.0                                   |
| RCM6       | 833.6                          | 623.7                                    | 9.9                          | 14.0                                   |
| RCM7       | 683.0                          | 623.1                                    | 9.6                          | 14.0                                   |
| RCM8       | 952.9                          | 623.3                                    | 10.9                         | 14.0                                   |
| RCM9       | 826.1                          | 623.5                                    | 9.5                          | 14.0                                   |

**Table 3: Classification of corrected RCMs according their reliability considering basic (mean, standard deviation and skew coefficient) and drought statistics (frequency, length, magnitude and intensity) for the meteorological and hydrological analyses. Lower numbers represent a higher reliability.**

| | Statistics used in the classification (basic and drought) | |
|---|---|---|
| | Meteorological | Hydrological |
| RCM1 | 4 | 3 |
| RCM2 | 2 | 2 |
| RCM3 | 9 | 6 |
| RCM4 | 6 | 8 |
| RCM5 | 7 | 7 |
| RCM6 | 5 | 9 |
| RCM7 | 3 | 5 |
| RCM8 | 8 | 4 |
| RCM9 | 1 | 1 |

**Table 4: Coeficient of variation of the historical and future series of precipitation, temperature, and streamflow generated from RCM2 and RCM9.**

| | Coefficient of variation (CV) | | |
|---|---|---|---|
| | Precipitation | Temperature | Streamflow |
| Historical | 0.80 | 0.46 | 0.69 |
| RCM2 | 1.07 | 0.41 | 0.84 |
| RCM9 | 1.10 | 0.42 | 1.07 |

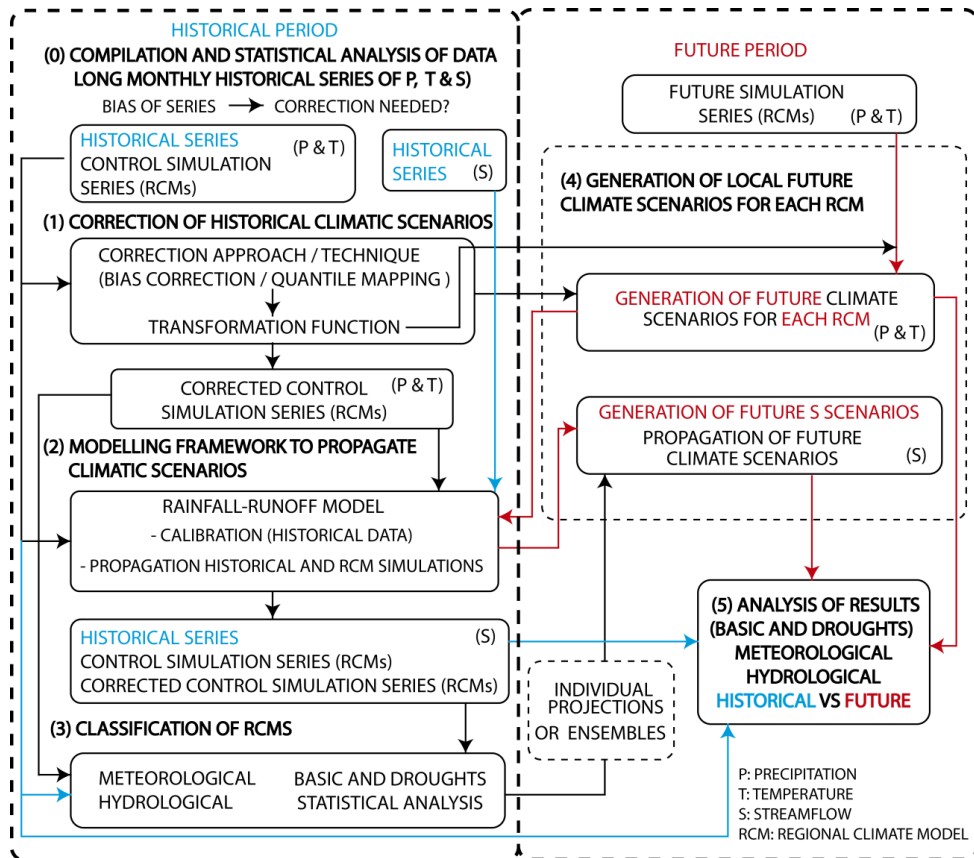

**Figure 1: Flow chart of the proposed methodology for the assessment of future meteorological and hydrological droughts.**

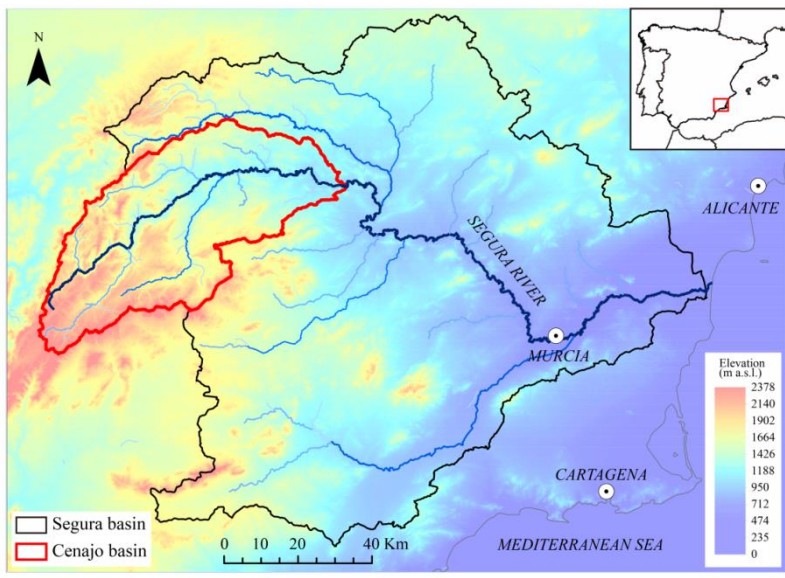

**Figure 2: Location of the case study.**

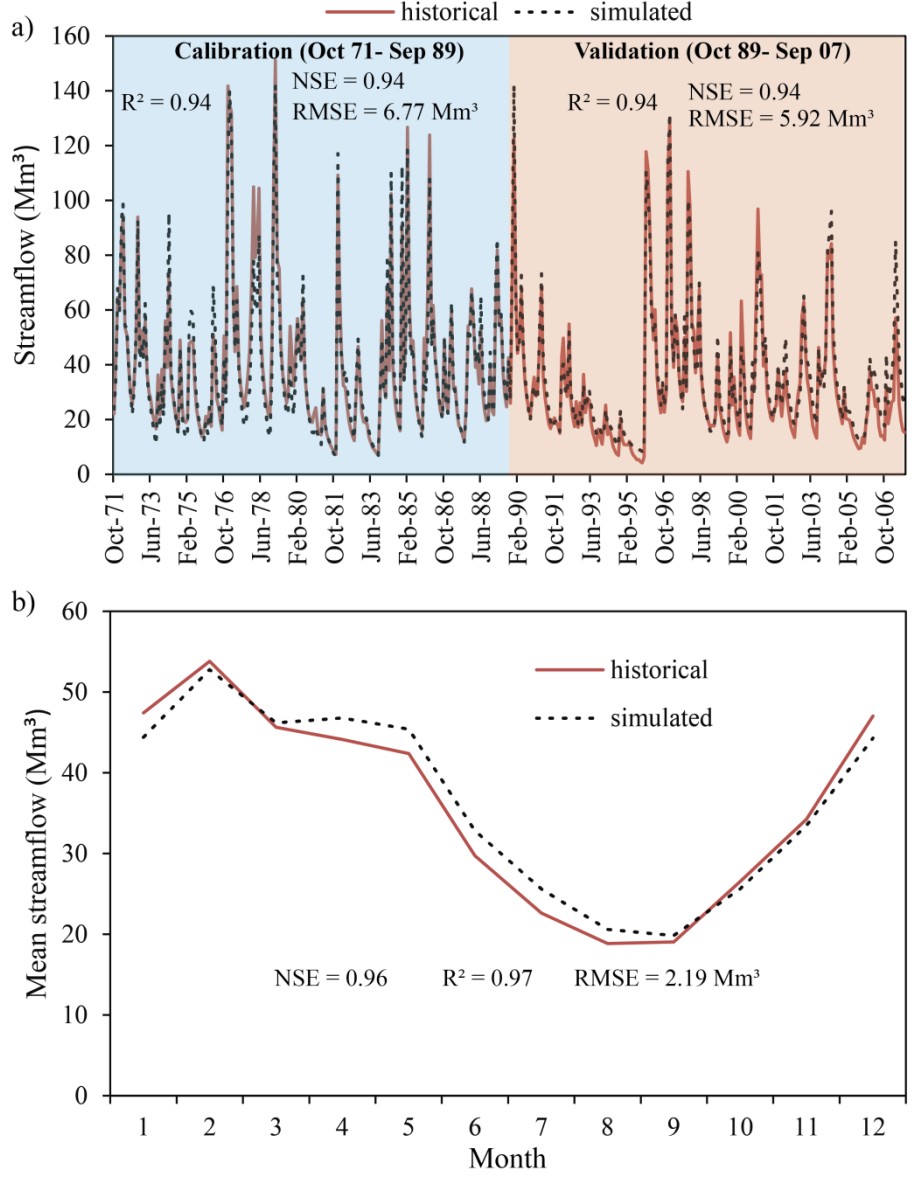


**Figure 3: Historical and simulated monthly streamflow series in the Cenajo basin for the calibration period (October 1971 to September 1989) and validation period (October 1989 to September 2007) (a) and mean monthly values within the mean year of the entire period (October 1971 to September 2007) (b).**

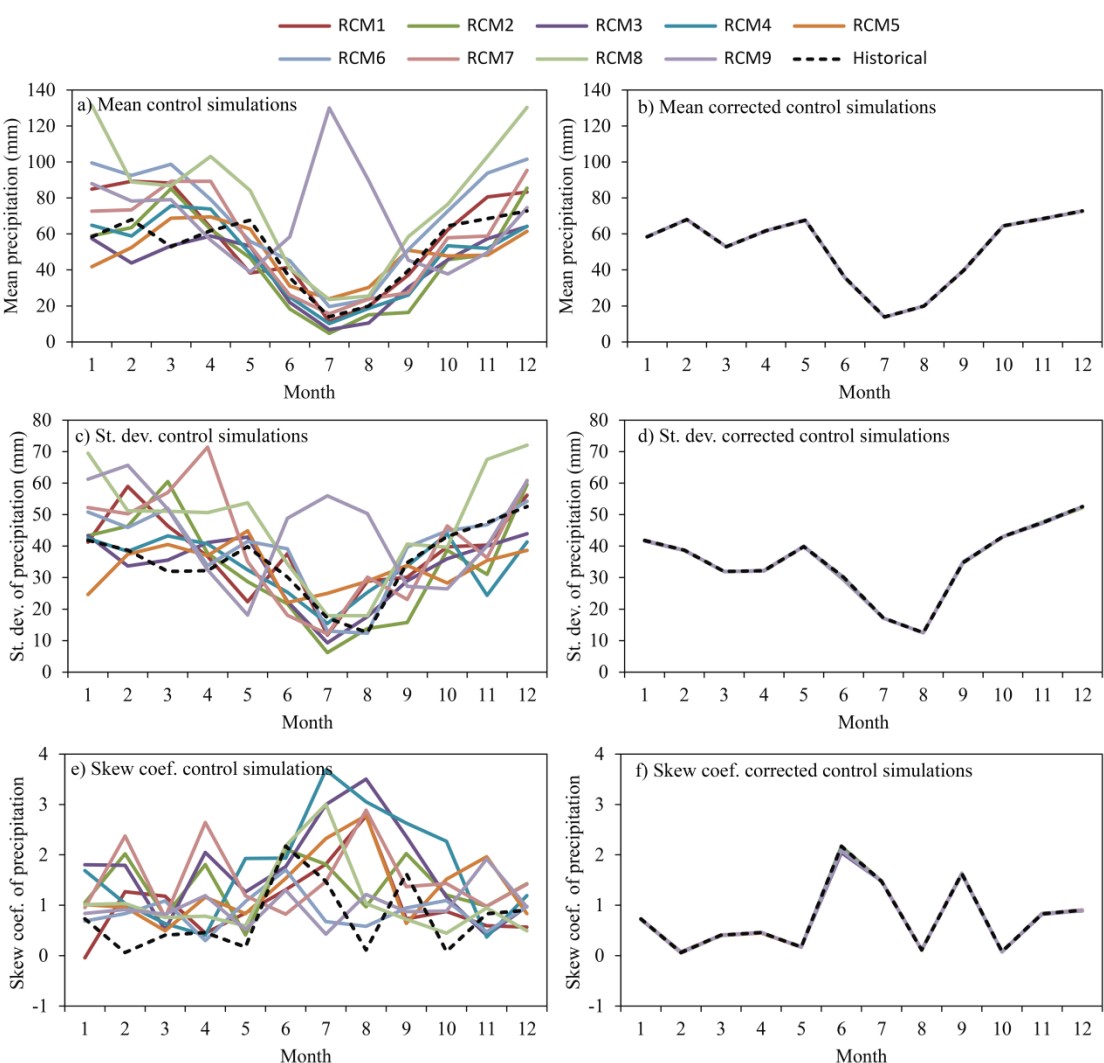

**Figure 4: Monthly mean, standard deviation, and skew coefficient of precipitation within the mean year of the period (1972-2001) for the historical and control simulation series (left column) and historical and corrected control simulation series (right column).**

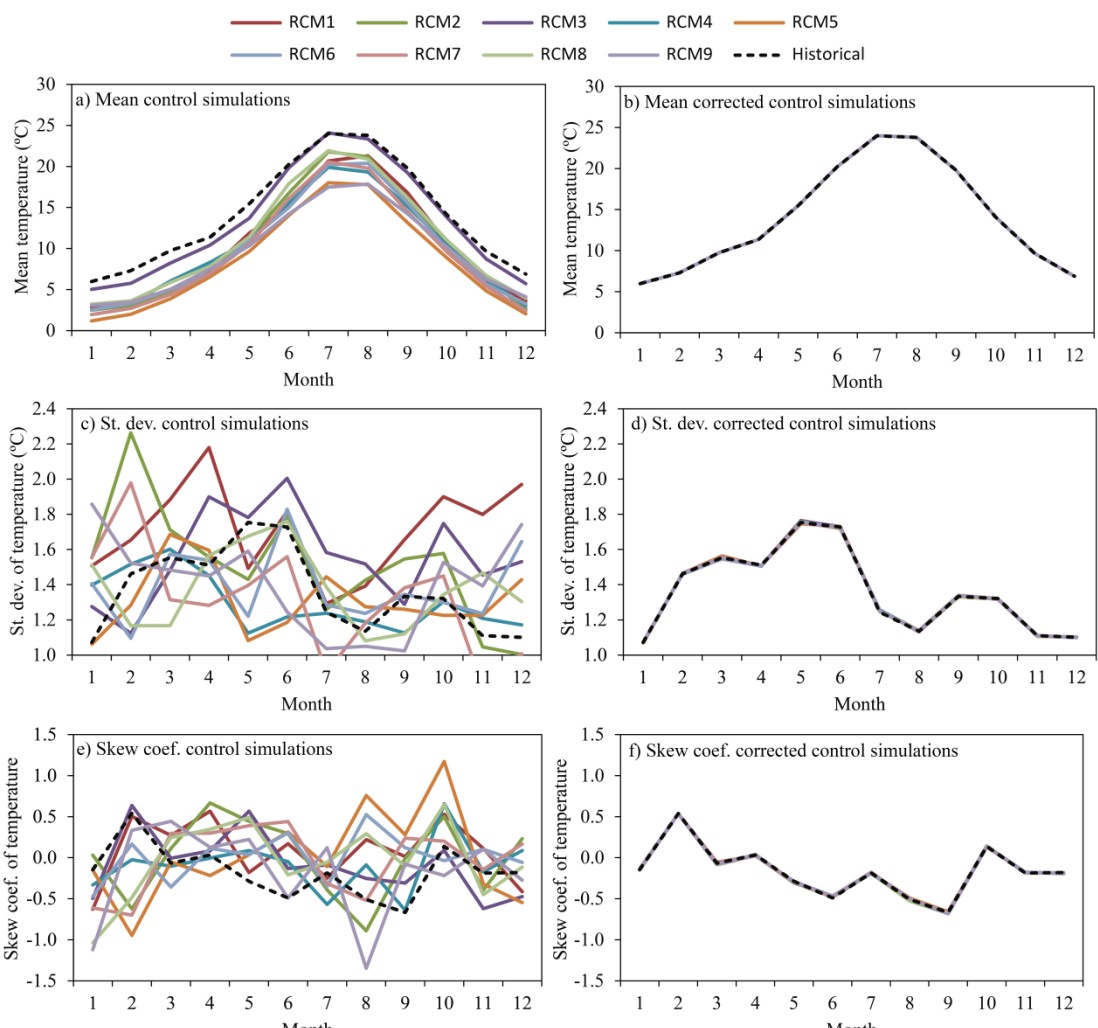

**Figure 5: Monthly mean temperature, standard deviation, and skew coefficient of temperature within the mean year of the period (1972-2001) for the historical and control simulation series (left column) and historical and corrected control simulation series (right column).**



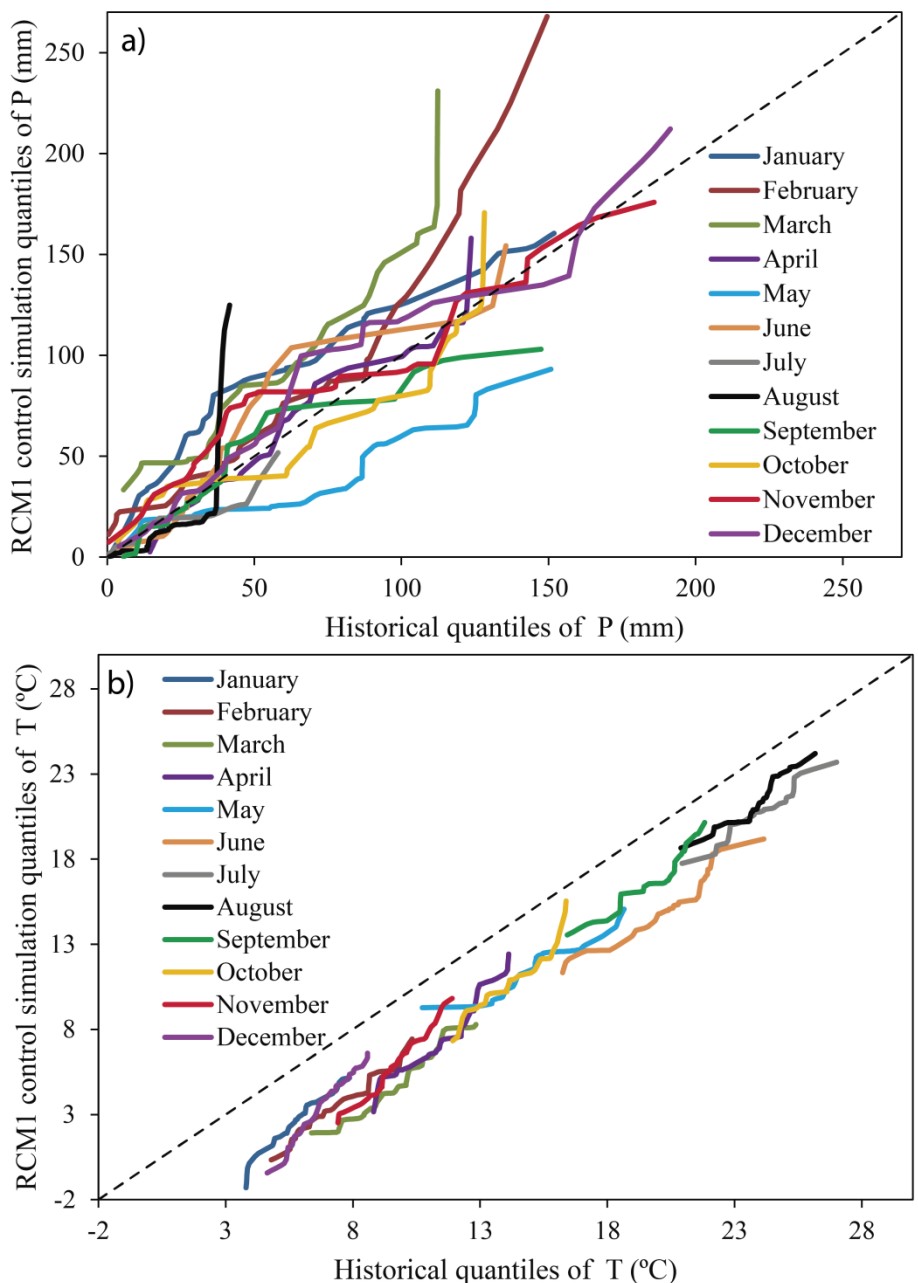

Figure 6: Precipitation and temperature quantiles of the observed and control series of the RCM1 simulations for each month of the year in the reference period (1972-2001).

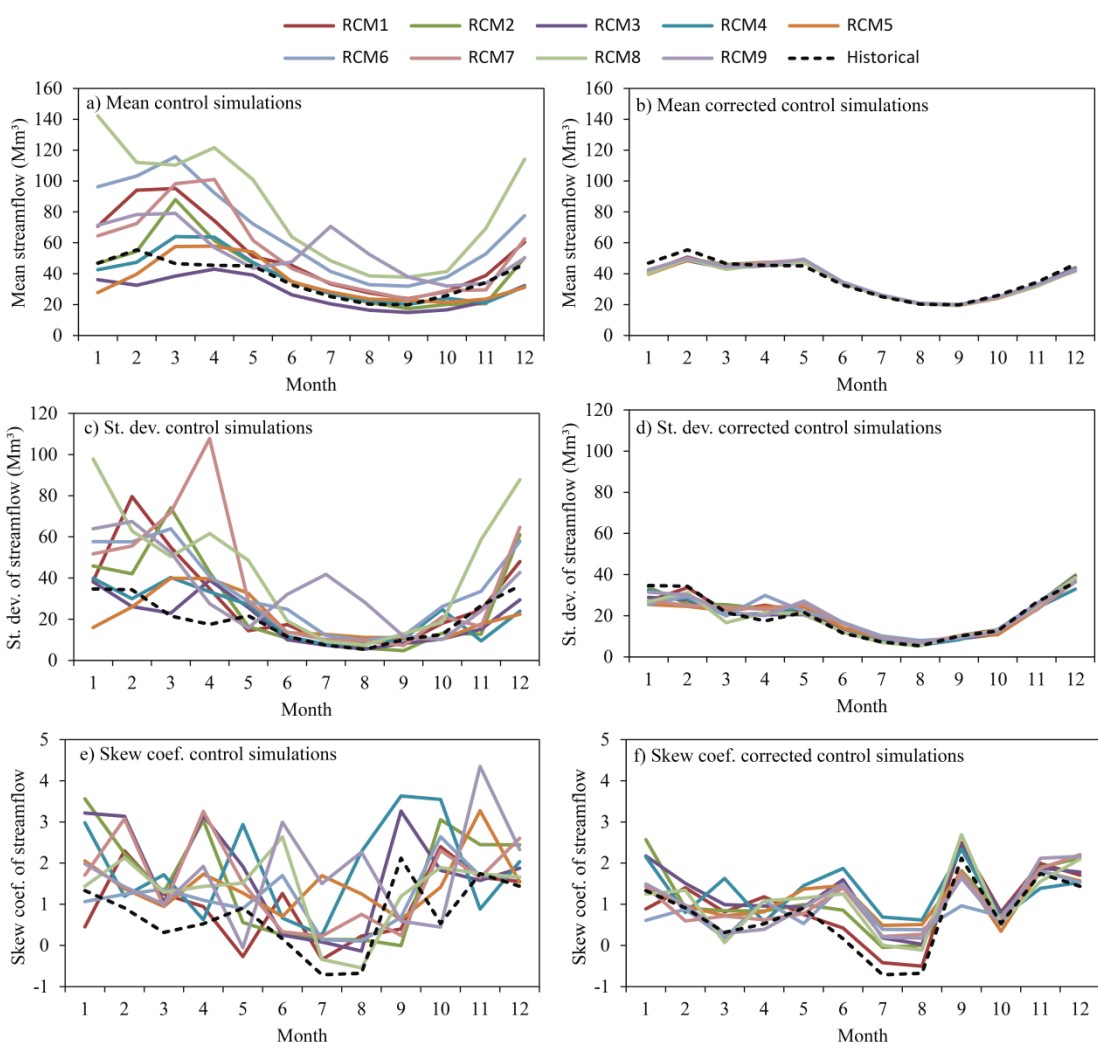

**Figure 7: Monthly mean, standard deviation, and skew coefficient of streamflow within the mean year of the period (1972-2001) for the historical and control simulation series (left column) and historical and corrected control simulation series (right column).**

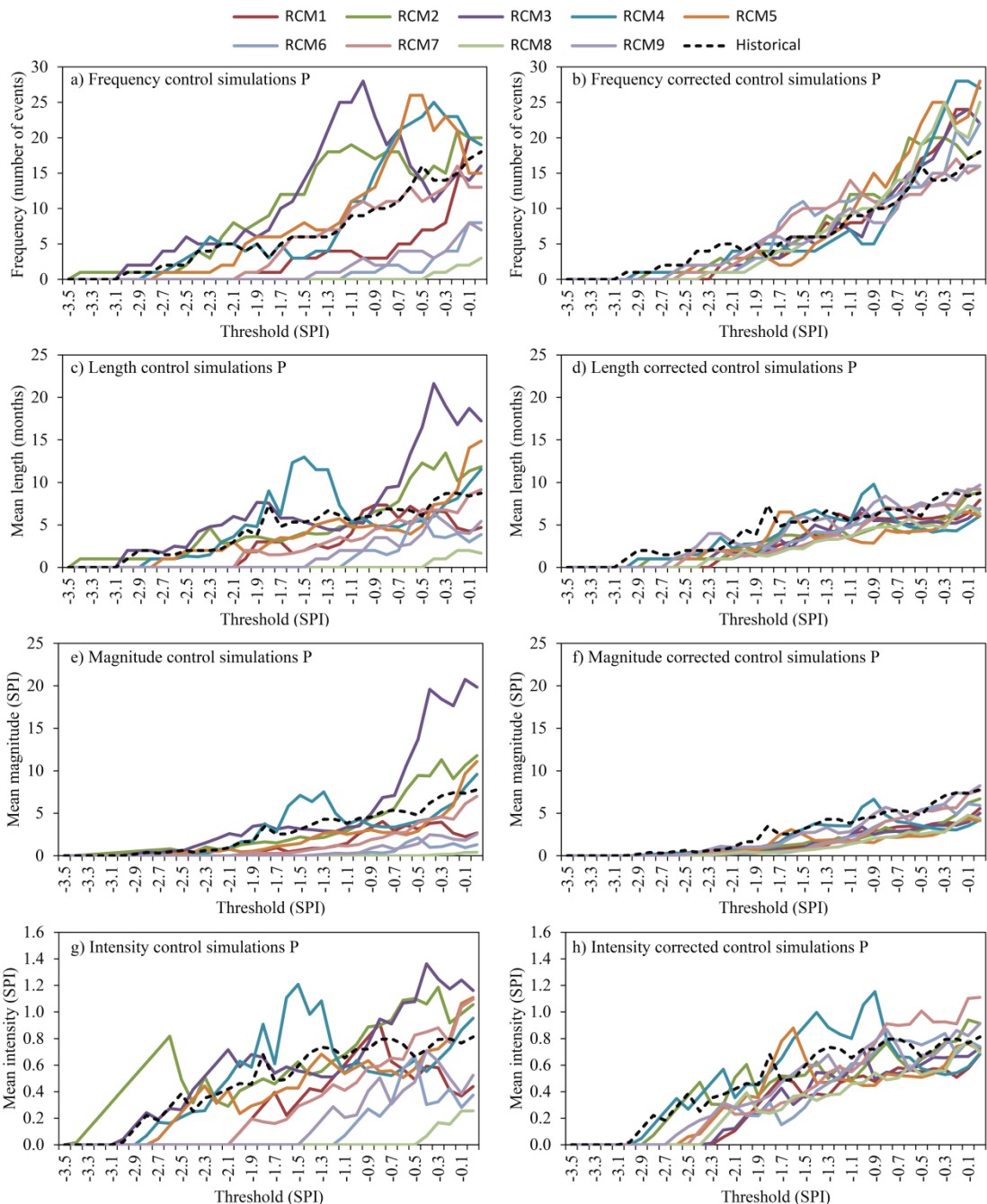

**Figure 8: Drought statistics (frequency, length, magnitude and intensity) of the period (1972-2001) for the historical and control simulation series (left column) and historical and corrected control simulation series (right column) for precipitation (meteorological droughts).**


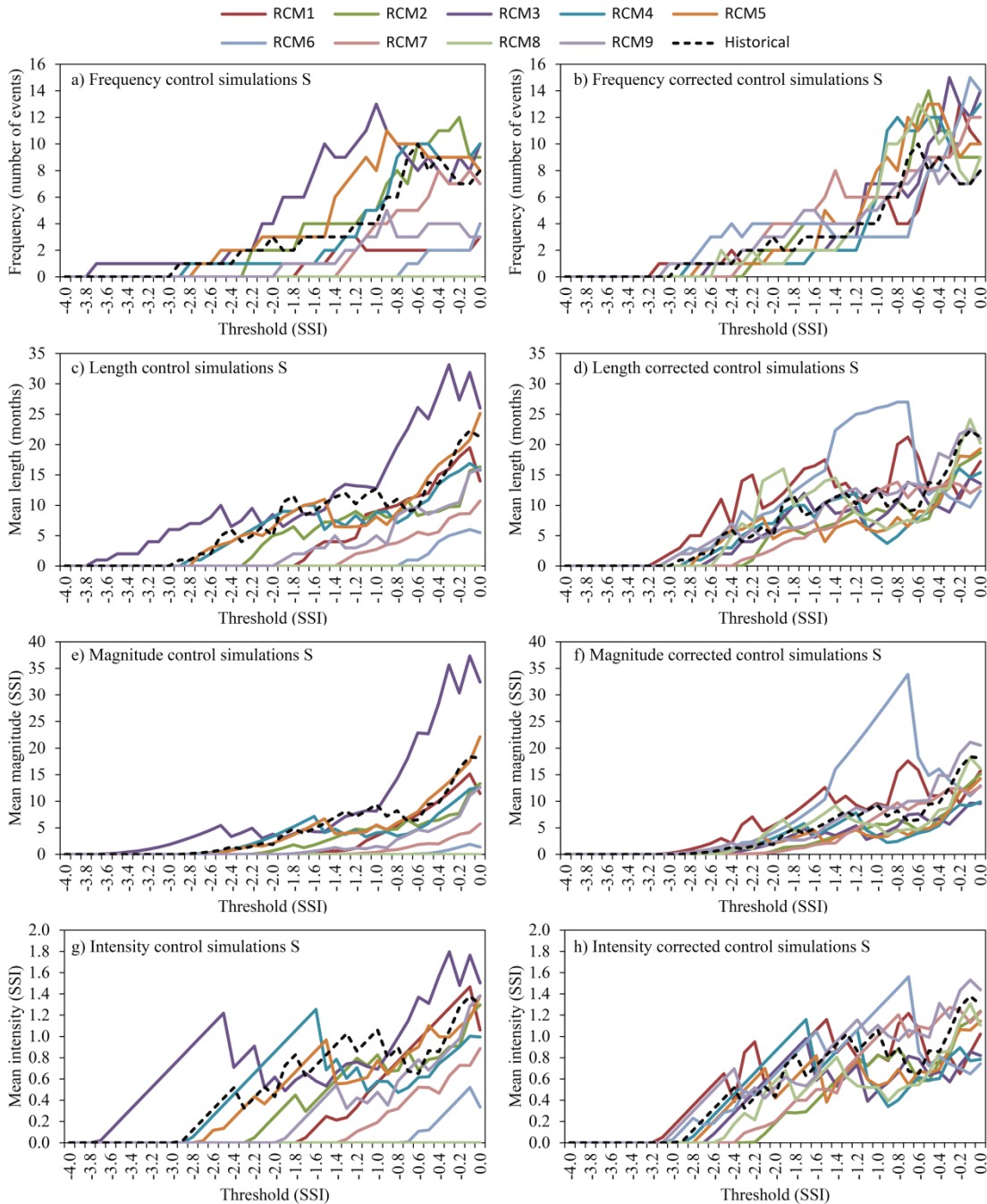

**Figure 9: Drought statistics (frequency, length, magnitude and intensity) of the period (1972-2001) for the historical and control simulation series (left column) and historical and corrected control simulation series (right column) for streamflow (hydrological droughts).**

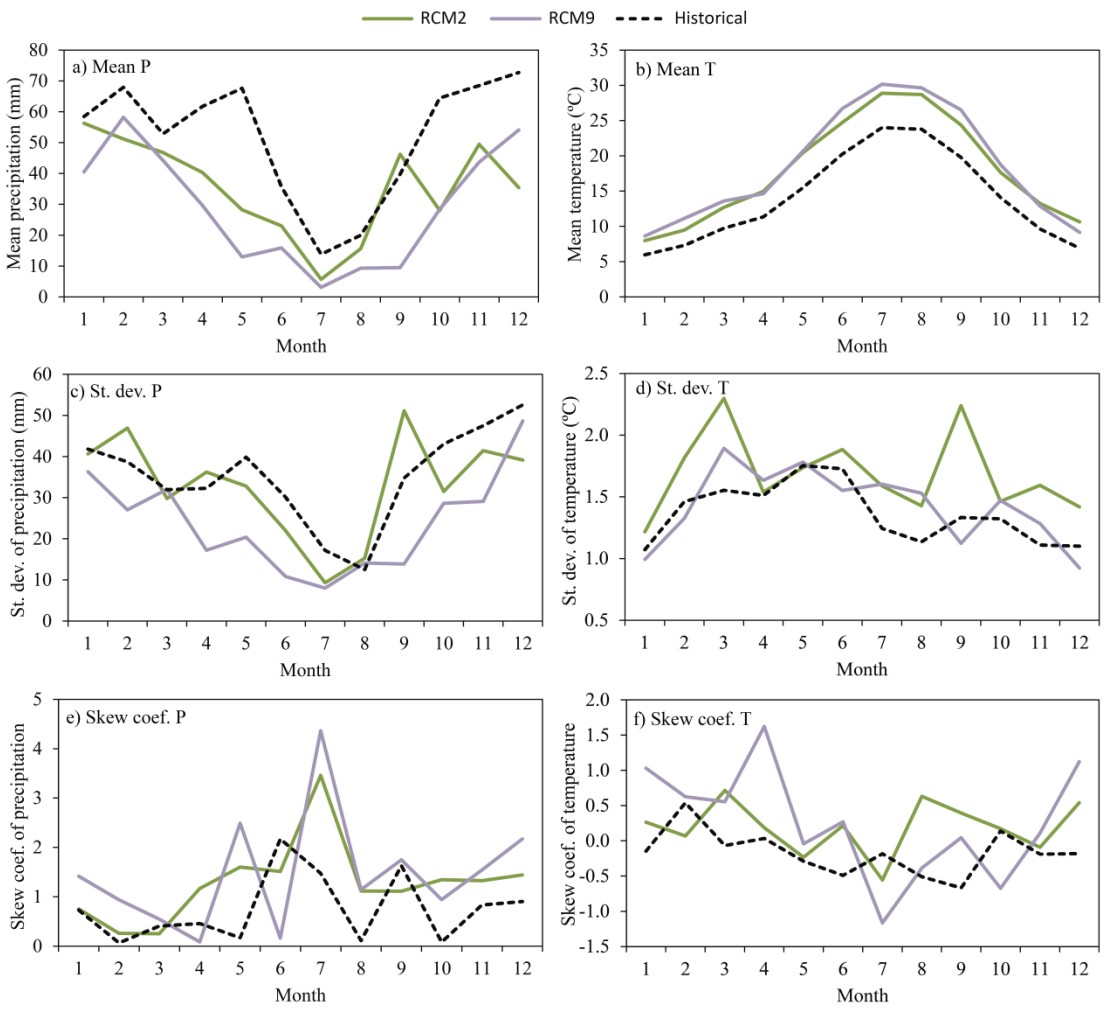


**Figure 10: Monthly mean, standard deviation, and skew coefficient within the mean year of the historical period (1972-2002) and future horizon (2071-2100) series (RCM 2 and 9) for precipitation (left column) and temperature (right column).**

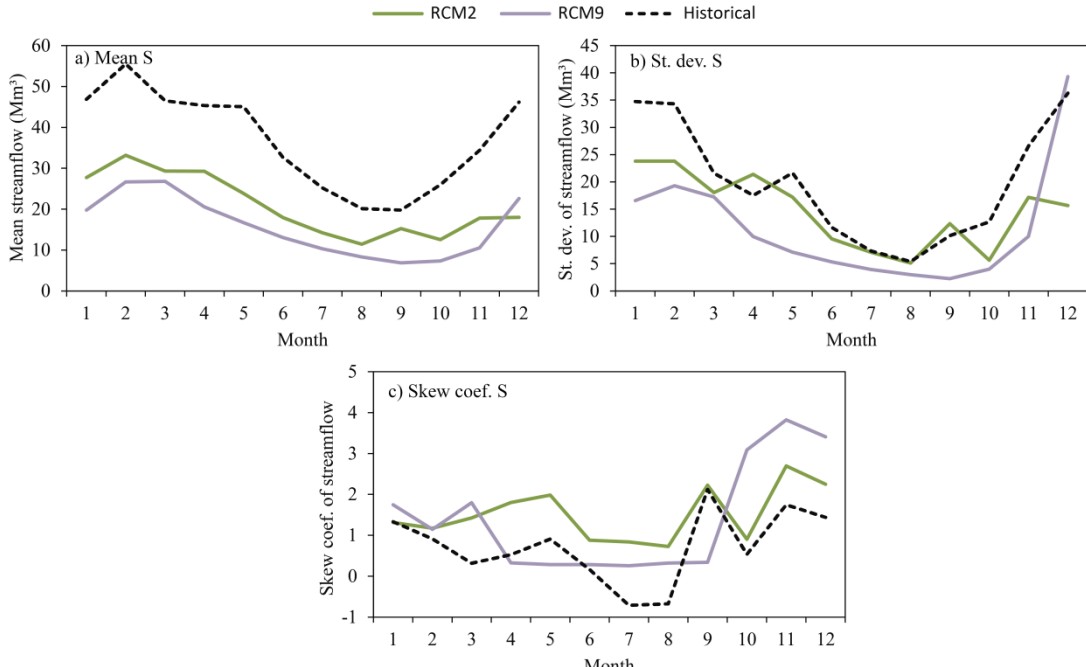

**Figure 11: Monthly mean (a), standard deviation (b), and skew coefficient (c) within the mean year of the historical period (1972-2002) and future horizon (2071-2100) series (RCM 2 and 9) for streamflow.**

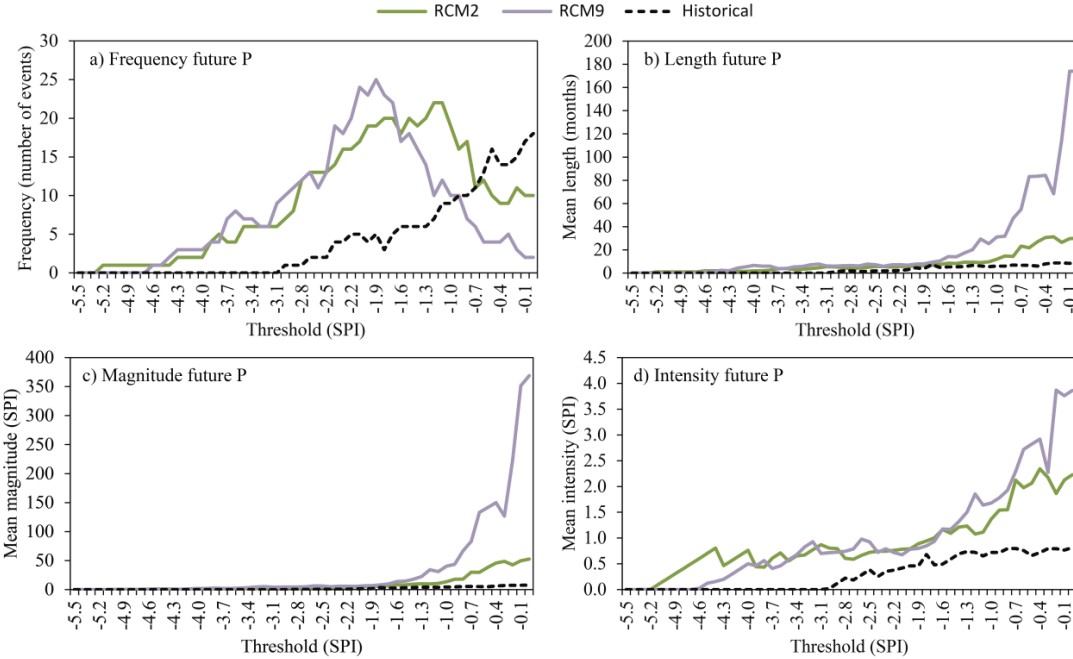

**Figure 12: Drought statistics [a) frequency, b) length, c) magnitude, d) intensity] of the historical period (1972-2002) and future horizon (2071-2100) series (RCM 2 and 9) for precipitation (meteorological droughts).**

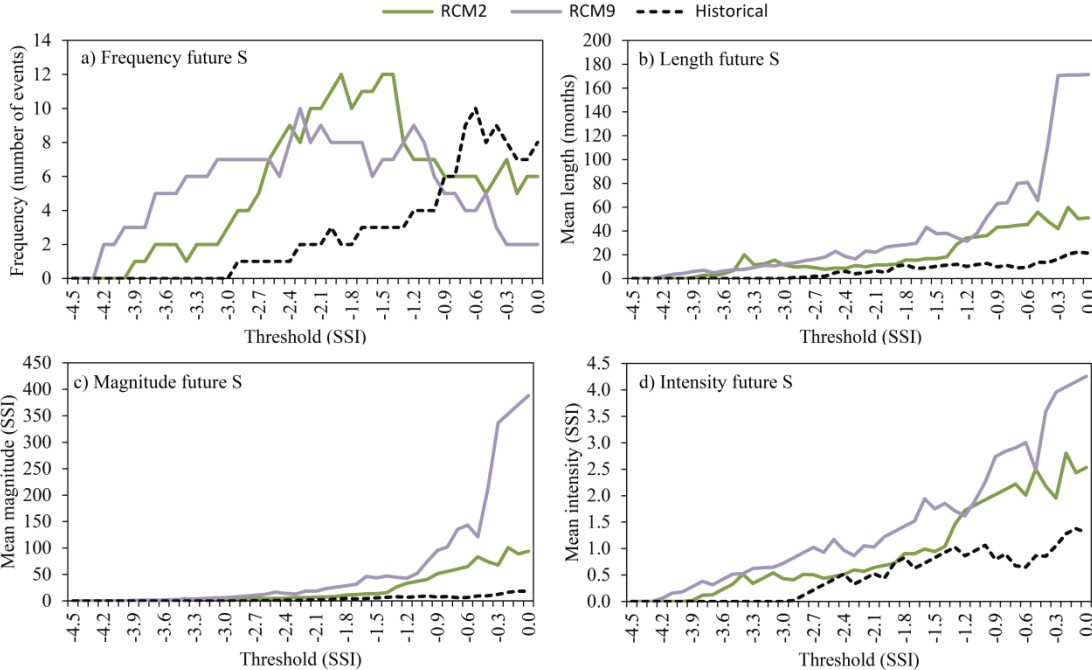

**Figure 13: Drought statistics [a) frequency, b) length, c) magnitude, d) intensity] of the historical period (1972-2002) and future horizon (2071-2100) series (RCM 2 and 9) for streamflow (hydrological droughts).**