# Peer review of "Do climate models that are better at approximating local meteorology also improve the assessment of hydrological responses? An analysis of basic and drought statistics."

_Natural Hazards and Earth System Sciences, 2021_

## Author Comment (AC1)

July 16$^{th}$, 2021

Dear Reviewer,

We are sending you the response to the referee comments (**nhess-2021-121-RC1**) of the manuscript -ref: **NHESS-2021-121-** and entitled "Do climate models that better approximate local meteorology improve the assessment of hydrological responses? Analyses of basic and drought statistics" by Antonio-Juan Collados-Lara, Juan-de-Dios Gómez-Gómez, David Pulido-Velazquez, and Eulogio Pardo-Igúzquiza.

We would like to express our sincere gratitude for your in-depth revision that will unquestionably help us to improve the manuscript. We have taken into account all the comments and we have provided response to them.

Thank you very much for your time and consideration.

Yours sincerely,

Antonio-Juan Collados-Lara, Juan-de-Dios Gómez-Gómez, David Pulido-Velazquez, and Eulogio Pardo-Igúzquiza.
Authors

*GENERAL COMMENTS*

*The authors present an assessment of the implications of bias correction methods for the assessment of the effect of climate change impacts on hydrological drought in a Mediterranean catchment. They applied a well-known bias correction method and chose to evaluate the performance of each RCM based not only on conventional statistics, but also on drought statistics. The authors discuss methodological issues related to the comparison of the RCM performance through the application of a rainfall-runoff model at the monthly time scale.*

*The topic is relevant for the audience of Natural Hazards and Earth System Science, the objectives are properly identified, the methodology for the analysis is adequate and the conclusions are relevant and correctly supported by the results and discussion. The overall organization of the manuscript is adequate, and it is clearly written. The analysis clearly shows the agreements and discrepancies between results obtained with different climatic forcings for the hydrologic model of choice. Therefore, I support publication of the work in Natural Hazards and Earth System Science.*

We thank the Reviewer for recognizing the positive aspects of our manuscript, the relevance of the topic and its interest for Natural Hazards and Earth System Science readership.

*SPECIFIC COMMENTS*

*I have several suggestions and comments, which I believe would improve the paper:*

*a) On section 4.2, the authors present their first assessment of discrepancy between historical observations and RCM control simulations. From Fig 4 and Fig 5, I gather that most models do a poor job at reproducing observed climate in the case study basin, particularly in seasonality of rainfall and temperature. I suggest adding a table with a comparison of mean annual values of precipitation and temperature to provide an objective comparison.*

Following the reviewer suggestion we have added the suggested table to the manuscript and we have also made references to it within the text.

Table 2: Mean annual values of precipitation and temperature for the historical and the RCM simulations (and corrected RCM simulations) in the reference period (1972-2001).

| | Mean annual precipitation (mm) | Mean annual corrected precipitation (mm) | Mean annual temperature (ºC) | Mean annual corrected temperature (ºC) |
|---|---|---|---|---|
| Historical | 623.6 | - | 14.0 | - |
| RCM1 | 700.5 | 623.5 | 10.4 | 14.0 |
| RCM2 | 550.7 | 623.1 | 10.4 | 14.0 |
| RCM3 | 503.6 | 623.3 | 13.2 | 14.0 |
| RCM4 | 571.7 | 623.6 | 10.1 | 14.0 |
| RCM5 | 588.7 | 623.3 | 8.5 | 14.0 |
| RCM6 | 833.6 | 623.7 | 9.9 | 14.0 |

| | | | | |
|---|---|---|---|---|
| RCM7 | 683.0 | 623.1 | 9.6 | 14.0 |
| RCM8 | 952.9 | 623.3 | 10.9 | 14.0 |
| RCM9 | 826.1 | 623.5 | 9.5 | 14.0 |

*b) The application of the quantile mapping technique is a critical step in the analysis. However, the authors do not provide much information on the procedure or the results while applied to the case study. There is a very brief introduction in the methodology section, with no details on how the original series are transformed. Regarding results, we can only see that bias for the three basic statistics has been eliminated. I think the authors should provide more information on the application of the technique to the case study and illustrate it with at least a figure showing the quantiles.*

We have included more information and references about quantile mapping technique within the methodology section:

The statistical transformation was defined by a quantile mapping technique based on empirical quantiles. We used the open-source R package qmap (Gudmundsson et al., 2012). Quantile mapping with empirical quantiles uses a non-parametric transformation function. In this approach the empirical cumulative distribution functions (CDFs) are approximated using tables of empirical quantiles. It estimates values of the empirical CDFs of observed and simulated time series for regularly spaced quantiles to create the table that relates observed and simulated time series (Enayati et al., 2021). The values between them are approximated by using linear interpolation. Accordingly, it uses interpolations to adjust a datum with unavailable quantile values. For each month of the year we used its table of empirical quantiles. These tables, which are obtained by using the CDF of the observed and simulated values (from RCMs), are also used to correct the future simulation (from RCMs). If the RCM values are larger than the historical ones used to estimate the empirical CDF, the correction found for the highest quantile of the historical period is used (Gudmundsson et al., 2012).

Following the reviewer suggestion we have also included a figure showing the precipitation and temperature quantiles for the observed and control simulation series obtained with the RCM1 for each month of the year in the reference period (1972-2001):

These differences force us to apply the correction approach defined in section 2.1 for all the RCMs considered. It uses the CDF (quantiles) of the historical series and control series obtained from the RCMs simulations to perform the correction. The precipitation and temperature quantiles of the observed and control simulation series of RCM1 in the reference period are showed in Fig. 6. The same information was generated for all the RCMs simulations and used to correct the RCMs outputs.

[Figure]

Figure 6: Precipitation and temperature quantiles of the observed and control series of the RCM1 simulations for each month of the year in the reference period (1972-2001).

*c) The authors chose to use SPI as drought index to characterize precipitation, but they should state the aggregation time step chosen in the analysis. The descriptive statistics used later in the paper (frequency, duration, magnitude, and intensity) should be formally introduced.*

We have included information about SPI aggregation time step and the used statistics of droughts:

The meteorological and hydrological drought analysis was developed by applying the Standard Precipitation index (SPI) (Bonaccorso et al., 2003; Livada and Assimakopoulos, 2007) and Standard Streamflow index (SSI) (Salimi et al., 2021),

respectively. They were estimated for periods of aggregation equal to 12 months. The calculation method requires the transformation of a gamma frequency distribution function to a normal standardized frequency distribution function. The statistics of the SPI/SSI series are obtained by applying the run theory (González and Valdés, 2006; Mishra et al., 2009) for different SPI/SSI thresholds from the lower SPI/SSI to 0. The frequency is defined as the number of droughts events for each SPI threshold. For each drought event, we assess its duration as the number of months that the SPI is below a given threshold, its magnitude as the summation of the SPI values for each month of the event, and its intensity as the minimum SPI value. For each threshold we estimate the mean duration, magnitude, and intensity as the mean values of the cited variables for all the drought events. The probability of occurrence of precipitation or streamflow for the SPI/SSI calculation, in the corrected control and future simulations, was obtained using the parameters calibrated from the observed series, in order to perform an appropriate comparison (Marcos-Garcia et al., 2017).

*d) I was a bit confused by the classification procedure. If I understood correctly, the RCM are assigned penalty values from 1 to 10 according to their ranking in each of 7 statistics. The final classification is obtained by averaging of the penalization for all statistics. However, the index chosen is divided by a normalizing value to allow comparisons across statistics. Why not directly use the index values instead of the penalties based on the ranking, to account for the relative deviations shown by each model?*

We propose these normalized values in order to give similar weight to all the statistics in the final classification. Note that the skew coefficient and droughts statistics have higher SE values. If we sum the SE values for all the statistics and we classify RCMs in accordance with it, the mean or standard deviation statistics will not influence in the final classification. It also allows us to define an index (SE) threshold below which the RCMs are not penalized. We have included it in the new version of the manuscript:

The penalization approach allows us to define an index (SE) threshold below which the RCMs are not penalized. It also allows us to give similar weight to all the statistics in the final classification. Note that the skew coefficient and droughts statistics have higher SE values. If we sum the SE values for all the statistics and we classify RCMs in accordance with it, the mean or standard deviation statistics will not influence in the final classification.

*e) On section 4.3, line 195, the authors state that there is a "correlation" between the order classification of corrected RCMs for meteorology and hydrology. By looking at Figure 9, I am not sure of this and I am afraid I must disagree. Figure 9 shows a scatter plot of nine values. The fitted regression line for the nine points has an R2 of 0.34, which is very low to conclude that there is a correlation (what is the significance level?). Even the blue line, which corresponds to only to 4 points, has a very low R2, of only 0.46. Finally, the authors should refrain from plotting the regression line for the two points corresponding to classification order <2, which obviously renders a perfect fit because there are only two points. By looking at the figure, I can also see an opposite "correlation" for the 5 points corresponding to classification order >4. The fitted regression line would have a negative slope, contradicting the initial statement. I think*

*this discussion should be reformulated. We all agree that good bias correction would improve the agreement between climate models and observations, but the authors need to provide objective results to draw this conclusion, which, by the way, is a central part of their contribution. I suggest separating the analysis of conventional statistics and drought statistics, since the bias correction procedure is specifically focused on fitting the results of climate models to observations and therefore one can expect (as shown in Figure 4 and Figure 5), that the index values are very low. This does not necessarily have to be the case for drought statistics, which are linked to the tail of the distribution. Perhaps showing the scatter plots of the actual index values obtained with all models would illustrate better the comparison of performance for meteorological and hydrological drought.*

The reviewer is right, the figure is confusing. When the Classification order <2, it obviously renders a perfect fit because there are only two points. For these two points we wanted to highlight that the first and second best models for both analyses (meteorological and hydrological analyses) are the same RCMs (RCM9 and RCM2). The third "best" model for meteorology is the fifth in the hydrology assessment, and the forth in meteorology the third in hydrology assessment. The results show that the best models for meteorology provides also the best results for hydrology, but as the Reviewer pointed we can see an opposite "correlation", if the analysis is extended to other models that are not the best ones. This discussion can be supported by Table 3, and for this reason, in accordance with the reviewer comment, we propose to eliminate Figure 9 in the new version of the manuscript. We have also modified section 4.3 to clarify it:

The classification of RCMs (after the bias correction of the simulations) based on the approximation of the meteorological and hydrological statistics (basic and drought statistics) by applying the procedure described in section 2.3 is included in Table 3. The two best corrected RCMs for meteorology (RCM2 and RCM9) are also the best models for hydrological assessment (maintain the first and second position in both cases). Nevertheless, the third "best" model for meteorology is the fifth in hydrological assessment, and the forth in meteorology the third in the hydrological assessment. Although they are still in the group of the best approaches, it demonstrates that there is not a cause-effect relationship; a better meteorological approximation not always means a better hydrological assessments. We only demonstrated that, in our case study, the RCMs that provide the best approximations of the meteorology provide the best assessments of hydrological impacts.

*TECHNICAL CORRECTIONS*

*From the formal standpoint, the paper is well written, correctly organized and adequately illustrated with tables and figures. I think the authors should rethink Figure 9 entirely.*

We thank the Reviewer for recognizing the positive aspects of the paper. Figure 9 was deleted because it was confusing (as the Reviewer pointed) and our findings are properly supported by Table 2.

*The authors should consider changing the term "asymmetry" coefficient for "skew" coefficient.*

We changed it along the text and figures.

*Page 4, line 119. I believe the normalizing value used in the denominator of equation 1 is useful for comparisons across statistics, not across RCMs, because the normalizing value (historical observations) is the same for all RCMs.*

The Reviewer is right. We modified the sentence:

Note that this index is a mean squared error of the corrected control with respects the historical values. It is divided by the square of the mean historical value in order to make the results comparable for different statistics.

*Figures 8 and 13. Please change SPI into SSI, since the plots refer to streamflow droughts.*

Done.

*Although I am not a native English speaker, I believe the following expression should be corrected:*

*On page 4, line 114, ... applying the "following" error index.*

Done:

We assessed the performance for each RCM in the reference period by applying the following error index (SE):

*On page 4, line 119, ... in order to make it comparable .*

Done:

Note that this index is a mean squared error of the corrected control with respects the historical values. It is divided by the square of the mean historical value in order to make the results comparable for different statistics.

*On page 6, line 174, ... when "they" compared different statistical techniques.*

Done:

It confirms the results obtained by Collados-Lara et al. (2018) when they compared different statistical correction techniques.

---

## Author Comment (AC2)

July 16$^{th}$, 2021

Dear Reviewer,

We are sending you the response to the referee comments (**nhess-2021-121-RC2**) of the manuscript -ref: **NHESS-2021-121-** and entitled "Do climate models that better approximate local meteorology improve the assessment of hydrological responses? Analyses of basic and drought statistics" by Antonio-Juan Collados-Lara, Juan-de-Dios Gómez-Gómez, David Pulido-Velazquez, and Eulogio Pardo-Igúzquiza.

We would like to express our sincere gratitude for your in-depth revision that will unquestionably help us to improve the manuscript. We have taken into account all the comments and we have provided response to them.

Thank you very much for your time and consideration.

Yours sincerely,

Antonio-Juan Collados-Lara, Juan-de-Dios Gómez-Gómez, David Pulido-Velazquez, and Eulogio Pardo-Igúzquiza.
Authors

*GENERAL COMMENTS*

*This study aims to provide further insights on the selection of Global Climate Models (GCMs) -Regional Climate Models (RCMs) combinations, according not only to their skills to reproduce the local climate during the selected historical period but also the local hydrology. Concretely, the authors calculate an error index for basic and drought statistics and use it to classify the GCMs-RCMs combinations according to their reliability for the assessment of meteorological and hydrological impacts. The selected methodology involves the bias correction of climate models' outputs through a quantile mapping (QM) approach based on empirical quantiles, the use of a lumped rainfall-runoff model to simulate monthly inflows from climate data and the use of standardized indices for drought characterization (namely the Standardized Precipitation Index (SPI) and the Standardized Streamflow Index (SSI)).*

*In my opinion, the paper addresses an important issue on the use of climate models' outputs for the assessment of climate change impacts at the basin scale. Besides, it is properly written and well presented.*

We thank the Reviewer for recognizing the importance of the issue and positive aspects of the manuscript.

*However, I miss a more critical approach to the potential shortcomings of the selected methodology, such as the underlying assumption of stationary bias, the impact of bias correction on the tails of the distribution (e.g. induced changes on the original climate change signal of the climate models), the pros and cons of pre-processing and post-processing the variables derived from climatic ones with regard to bias correction (e.g. performance of the hydrological model simulations over the validation period) or the potential effects of neglecting the inter-variable dependence of climate variables (e.g use of univariate bias correction methods against multivariable ones or ignoring the role of temperature in drought onset) on the assessment of climate change impacts on water resources.*

The reviewer is right; the shortcomings of the selected methodology should be stated. We included a new section in Discussion to point the limitations and future works related to the proposed approach:

5.1 Hypothesis assumed, limitations and future works

Although we have demonstrated the utility of the proposed approach to assess future impacts on meteorological and hydrological droughts, we want to highlight some hypothesis and limitations assumed and to identify potential future research aligned with this study:

- We have used a bias correction method based on the assumption of bias stationarity of climate model outputs. However, this assumption may not be valid to study some problems due to the significance of the influence of climate variability on them. Other approaches should be explored to take into account non-stationarity bias of RCMs simulations (e.g. Hui et al., 2020).
- We have applied the same bias correction procedure for all the range values in accordance with the climatic variable distribution function. We did not consider

the impact of bias correction techniques on the tails of the distribution, which could be important to analyse extremes (Volosciuk et al., 2017).

- In this work a univariate bias correction method is used. It does not consider the dependence between precipitation and temperature which could be explored in future assessments. Meyer et al. (2019) found that incorporating or ignoring inter-variable relationships between temperature and precipitation could impact the conclusions drawn in hydrological climate change impact studies in alpine catchments.

- The streamflow information available for this case study cannot be divided into two long-enough (e.g. 30 years) series representative of the climate/hydrology to perform explicitly a validation of the bias correction models (Chen et al., 2021). We assumed that the statistics of any long-enough periods remain invariant. In this case the calibration implicitly could be considered validated, due to the fact that the same results would be obtained under this hypothesis for any other period representatives of the climate/hydrology conditions.

- In our case study the influence of temperature was considered only in the hydrological assessment by using rainfall-runoff models. However other meteorological droughts indices that consider temperature could be included in the analysis [e.g. the Standardised Precipitation-Evapotranspiration Index (SPEI) (García-Valdecasas Ojeda et al., 2021)].

- The corrected control simulation series obtained by using a quantile mapping bias correction presents a very good performance with respect the historical series in terms of basic statistics. In the case of droughts (calculated from SPI/SSI) the bias correction approach clearly improves the fit of the RCM simulation series to the historical series, but the performance is lower than for basic statistics. Other bias correction procedures should be explored to improve the performance for droughts statistics.

- The proposed method has not been tested in other typologies of basin, as for example in Alpine basins where snowmelt component may have a significant influence on the results.

*SPECIFIC COMMENTS*

*Lines 27-29: "For instance, we have Palmer Drought Severity Index (...)". I will also mention the Standardized Precipitation Evapotranspiration Index (SPEI, Vicente-Serrano et al., 2010).*

*Vicente-Serrano S.M., Santiago Beguería, Juan I. López-Moreno, (2010) A Multi-scalar drought index sensitive to global warming: The Standardized Precipitation Evapotranspiration Index - SPEI. Journal of Climate 23: 1696-1718.*

Done:

For instance, we have Palmer Drought Severity Index (PDSI) (Palmer, 1965), Crop Moisture Index (CMI) (Palmer, 1968), Standardized Precipitation Index (SPI) (McKee et al., 1993), Soil Moisture Drought Index (SMDI) (Hollinger et al., 1993), Vegetation Condition Index (VCI) (Liu and Kogan, 1996), Standardized Precipitation Evapotranspiration Index (SPEI) (Vicente-Serrano et al., 2010).

*Lines 91-92: "This is the reason that justifies the selection of quantile mapping (using empirical quantiles) for this study". Have the effects of inter-variable dependence been considered before selecting an univariate bias correction method? In the case of alpine catchments, Meyer et al. (2019) found that incorporating or ignoring inter-variable relationships between air temperature and precipitation data could impact the conclusions drawn in hydrological climate change impact studies.*

*Meyer, J., Kohn, I., Stahl, K., Hakala, K., Seibert, J., Cannon, A. J. (2019). Effects of univariate and multivariate bias correction on hydrological impact projections in alpine catchments. Hydrology and Earth System Sciences, 23, 3, 1339-1354, https://hess.copernicus.org/articles/23/1339/2019/*

We agree with the reviewer. We did not consider dependence between precipitation and temperature, which may produce significant impacts in some basins, as alpine catchments. Our case study is not located in an alpine catchment, but we have also added it as a potential limitation of the methodology:

In this work a univariate bias correction method is used. It does not consider the dependence between precipitation and temperature which could be explored in future assessments. Meyer et al. (2019) found that incorporating or ignoring inter-variable relationships between temperature and precipitation could impact the conclusions drawn in hydrological climate change impact studies in alpine catchments.

*Lines 105-109: "The meteorological drought analysis was developed by applying the Standard Precipitation index (SPI)". What about the role of temperature? As multiple authors have already pointed out, SPEI usually shows more severe increases in future drought events than those from SPI (e.g. García-Valdecasas Ojeda et al., 2021) and therefore I recommend to include it in the analysis. Which aggregation periods, statistical distributions and thresholds are considered for both the SPI and the SSI?*

*García-Valdecasas Ojeda, M., Gámiz-Fortis, S.R., Romero-Jiménez, E., Rosa-Cánovas, J.J., Yeste, P., Castro-Díez, Y., Esteban-Parra, M.J. (2021). Projected changes in the Iberian Peninsula drought characteristics, Science of The Total Environment, Volume 757, 143702, ISSN 0048-9697, https://doi.org/10.1016/j.scitotenv.2020.143702.*

In our case study the temperature was considered to propagate climate impacts on hydrology by using the rainfall-runoff model. We also included the SPEI as a possible index to be studied in future works:

In our case study the influence of temperature was considered only in the hydrological assessment by using rainfall-runoff models. However other meteorological droughts indices that consider temperature could be included in the analysis [e.g. the Standardised Precipitation-Evapotranspiration Index (SPEI) (García-Valdecasas Ojeda et al., 2021)].

We have also specified within the new version of the manuscript the aggregation periods, statistical distributions and thresholds considered for SPI and SSI:

The meteorological and hydrological drought analysis was developed by applying the Standard Precipitation index (SPI) (Bonaccorso et al., 2003; Livada and Assimakopoulos, 2007) and Standard Streamflow index (SSI) (Salimi et al., 2021),

respectively. They were estimated for periods of aggregation equal to 12 months. The calculation method requires the transformation of a gamma frequency distribution function to a normal standardized frequency distribution function. The statistics of the SPI/SSI series are obtained by applying the run theory (González and Valdés, 2006; Mishra et al., 2009) for different SPI/SSI thresholds from the lower SPI/SSI to 0. The frequency is defined as the number of droughts events for each SPI threshold. For each drought event, we assess its duration as the number of months that the SPI is below a given threshold, its magnitude as the summation of the SPI values for each month of the event, and its intensity as the minimum SPI value. For each threshold we estimate the mean duration, magnitude, and intensity as the mean values of the cited variables for all the drought events. The probability of occurrence of precipitation or streamflow for the SPI/SSI calculation, in the corrected control and future simulations, was obtained using the parameters calibrated from the observed series, in order to perform an appropriate comparison (Marcos-Garcia et al., 2017).

*Lines 136-146: In my opinion, the climate and hydrological regime of the Cenajo basin should be properly characterized in the Case study section.*

Done:

The Cenajo basin has a Mediterranean climate. In the period 1972-2001, the mean annual precipitation was 623.6 mm and the mean temperature 14.0 ℃. In the same period the mean annual streamflow was 443.6 Mm³. This is a critical area where climate change will exacerbate these problems by reducing the availability of resources and increasing irrigation requirements. It will also cause an increase in the magnitude and frequency of extreme events, such as droughts.

*Line 149: "CORDEX project (2013)". Reference?*

Added:

The RCMs were retrieved from the CORDEX project (CORDEX PROJECT, 2013), with a spatial resolution of 0.11º (approximately 12.5 km).

*Lines 151-152: "We also used official monthly natural streamflow data within the Cenajo basin for the historical period 1972 -2001 (adopted as reference)". This reference period is not consistent with the calibration period of the rainfall-runoff model (October 1971 to September 2007, line 157).*

Yes, the periods are different. We used the available historical information to calibrate the rainfall-runoff model and selected a 30-year period, which is usually used to perform the climatic change analysis. We have clarified it within the manuscript:

The rainfall-runoff model for the Cenajo basin was calibrated and validated using the available monthly climatic data (precipitation, temperature, and potential evapotranspiration) and streamflow data for the period October 1971 to September 2007. We split the period with available data in two to perform a calibration (from October 1971 to September 1989) and validation (October 1989 to September 2007) of the model. The performance of the model was assessed by using the Nash-Sutcliffe efficiency (NSE) coefficient, the correlation coefficient (R2), and the root mean squared error (RMSE). These statistics and the historical and simulated streamflow series are

showed in Fig. 3a. For the entire period (October 1971 to September 2007) the performance is also good (NSE = 0.94) and it is higher (NSE = 0.96) if we focus on the monthly mean within the mean year for the entire period (Fig. 3b). The model was used to propagate the impacts of climatic variables on streamflow between 1972 and 2001, a 30 year horizon, which is a period of time usually used in climate change analysis.

*Line 153: "(...) Spanish Ministry of Agriculture, food and environment". The competences of this former ministry have been assumed by the current Ministry for the Ecological Transition and the Demographic Challenge.*

Thank you. We updated it:

We also used official monthly natural streamflow data within the Cenajo basin for the historical period 1972-2001 (adopted as reference). These data were taken from the available information coming from the Spanish Ministry for the Ecological Transition and the Demographic Challenge.

*Lines 156-161: What is the validation period? Goodness of fit for the validation period?*

Thank you to the reviewer we realized that we did not explain properly this paragraph. We only presented results for the entire historical period. In the new version we included information about calibration and validation periods and its performance statistics. We also modified Figure 3:

The rainfall-runoff model for the Cenajo basin was calibrated and validated using the available monthly climatic data (precipitation, temperature, and potential evapotranspiration) and streamflow data for the period October 1971 to September 2007. We split the period with available data in two to perform a calibration (from October 1971 to September 1989) and validation (October 1989 to September 2007) of the model. The performance of the model was assessed by using the Nash-Sutcliffe efficiency (NSE) coefficient, the correlation coefficient (R2), and the root mean squared error (RMSE). These statistics and the historical and simulated streamflow series are showed in Fig. 3a. For the entire period (October 1971 to September 2007) the performance is also good (NSE = 0.94) and it is higher (NSE = 0.96) if we focus on the monthly mean within the mean year for the entire period (Fig. 3b). The model was used to propagate the impacts of climatic variables on streamflow between 1972 and 2001, a 30 year horizon, which is a period of time usually used in climate change analysis.

[Figure]

Figure 3: Historical and simulated monthly streamflow series in the Cenajo basin for the calibration period (October 1971 to September 1989) and validation period (October 1989 to September 2007) (a) and mean monthly values within the mean year of the entire period (October 1971 to September 2007) (b).

*Lines 178-180: "The fit of the corrected control simulation series of streamflow to the historical series is not as good as for precipitation and temperature, but a remarkable improvement is observed". What could be the reasons for this?*

The reason could be that we are neglecting the inter-variable dependence of climate variables not considering the dependence between precipitation and temperature when the bias correction is applied. Therefore, some differences might appear in the streamflow that depend on the combined interaction of both variables. We have included it within the new version of the manuscript:

The fit of the corrected control simulation series of streamflow to the historical series is not as good as for precipitation and temperature, but a remarkable improvement is observed. The reason could be that we are neglecting the inter-variable dependence of

climate variables not considering the dependence between precipitation and temperature when the bias correction is applied. Therefore, some differences might appear in the streamflow that depend on the combined interaction of both variables.

We have also commented it in the new subsection 5.1 Hypothesis assumed, limitations and future works (line xx-yy of the new version of the manuscript):

In this work a univariate bias correction method is used. It does not consider the dependence between precipitation and temperature which could be explored in future assessments. Meyer et al. (2019) found that incorporating or ignoring inter-variable relationships between temperature and precipitation could impact the conclusions drawn in hydrological climate change impact studies in alpine catchments.

*What about the performance over the validation period? (e.g. see Chen et al., 2021).*

*Chen, J., Arsenault, R., Brissette, F. P., & Zhang, S. (2021). Climate change impact studies: Should we bias correct climate model outputs or post-process impact model outputs? Water Resources Research, 57, e2020WR028638. https://doi.org/10.1029/2020WR028638*

The available streamflow information cannot be divided into two long-enough (e.g. 30 years) series representative of the climate/hydrology whose statistics are nearly invariant, we cannot perform, explicitly, a validation of the correction model. We assumed that the statistics of any long-enough periods remain invariant. In this case the calibration implicitly could be considered validated, due to the fact that the same results would be obtained under this hypothesis for any other period representatives of the climate/hydrology conditions. We stated it as a hypothesis and limitation assumed in our approach:

The streamflow information available for this case study cannot be divided into two long-enough (e.g. 30 years) series representative of the climate/hydrology to perform explicitly a validation of the bias correction models (Chen et al., 2021). We assumed that the statistics of any long-enough periods remain invariant. In this case the calibration implicitly could be considered validated, due to the fact that the same results would be obtained under this hypothesis for any other period representatives of the climate/hydrology conditions.

*Line 189: "Note that in this case we refer to the Standard Streamflow Index (SSI)". This index should be properly defined previously (in the Methodology section), along with an appropriate reference.*

Done:

The meteorological and hydrological drought analysis was developed by applying the Standard Precipitation index (SPI) (Bonaccorso et al., 2003; Livada and Assimakopoulos, 2007) and Standard Streamflow index (SSI) (Salimi et al., 2021), respectively. They were estimated for periods of aggregation equal to 12 months. The calculation method requires the transformation of a gamma frequency distribution function to a normal standardized frequency distribution function. The statistics of the SPI/SSI series are obtained by applying the run theory (González and Valdés, 2006; Mishra et al., 2009) for different SPI/SSI thresholds from the lower SPI/SSI to 0.

*Lines 198-199: "Therefore, we demonstrated that RCMs that allow better approximations of the meteorology provide better assessments of hydrological impacts". Although it seems quite straightforward (as rainfall-runoff models require climatic variables as inputs), I think that this statement should be carefully discussed before generalizing it: would it hold true if basins with very different hydrological regimes were considered? (e.g. important groundwater or snowmelt components?).*

The Reviewer is right, we have modified this paragraph:

The classification of RCMs (after the bias correction of the simulations) based on the approximation of the meteorological and hydrological statistics (basic and drought statistics) by applying the procedure described in section 2.3 is included in Table 3. The two best corrected RCMs for meteorology (RCM2 and RCM9) are also the best models for hydrological assessment (maintain the first and second position in both cases). Nevertheless, the third "best" model for meteorology is the fifth in hydrological assessment, and the forth in meteorology the third in the hydrological assessment. Although they are still in the group of the best approaches, it demonstrates that there is not a cause-effect relationship; a better meteorological approximation not always means a better hydrological assessments. We only demonstrated that, in our case study, the RCMs that provide the best approximations of the meteorology provide the best assessments of hydrological impacts.

We also pointed in the section "5.1 Hypothesis assumed, limitations and future works" new version the interest of consider basins with different hydrological regimes to test the proposed method:

The proposed method has not been tested in other typologies of basin, as for example in Alpine basins where snowmelt component may have a significant influence on the results.

*Lines 208-209: "Both RCMs predicts a decrease of the variability in precipitation and an increase of the variability of temperature in the future". This is an interesting result, as precipitation variability is generally expected to increase in a climate change context (e.g. Pendergrass et al., 2017). Concretely, for the Mediterranean regions, Polade et al. (2017) concluded that a decrease in the frequency of daily precipitation events, combined with an increase in the amount of precipitation delivered in relatively rare heavy events, yielded greater year-to-year variability in total precipitation. In my opinion, this result should be discussed in the context of existing literature on future climate variability in the Mediterranean area. Which could be the potential role of bias correction in this result? For example, Maraun (2013) investigated the role of bias correction in modifying relative trends in annual precipitation maxima from a RCM and found that the RCM underestimated observed variability, which led to substantial amplification by quantile mapping of modeled trends in extremes. Besides, it would be interesting to examine the future trends obtained from the rest of the GCM/RCM combinations.*

*Maraun, D. (2013). Bias correction, quantile mapping, and downscaling: Revisiting the inflation issue. J. Climate,26,2137–2143, doi:10.1175/JCLI-D-12-00821.1*

*Pendergrass, A.G., Knutti, R., Lehner, F. et al. Precipitation variability increases in a warmer climate. Sci Rep 7, 17966 (2017). https://doi.org/10.1038/s41598-017-17966-y*

*Polade, S. D., Gershunov, A., Cayan, D. R., Dettinger, M. D., & Pierce, D. W. (2017). Precipitation in a warming world: Assessing projected hydro-climate changes in California and other Mediterranean climate regions. Scientific reports, 7(1), 10783. https://doi.org/10.1038/s41598-017-11285-y*

Thank you to the Reviewer comments we realized that this sentence was wrong. Both RCMs forecast a decrease of the standard deviation of precipitation. It is not equivalent to the variability. We have calculated the coefficient of variation of the historical and future series obtained with RCM2 and RCM9 and we obtained 0.80, 1.07, and 1.10 respectively. Therefore, the variability is higher in the future. We have modified this sentence and included the references suggested by the Reviewer.

The considered RCMs predict significant reductions of mean precipitation (-31.6 % and -44.0 % for RCM2 and RCM9 respectively) and increase of mean temperature (26.0 % and 32.2 % for RCM2 and RCM9 respectively) (see Fig. 10a and 10b respectively). The average change in monthly standard deviation of precipitation is -6.2 % and -32.3 % for RCM2 and RCM9 respectively. In the case of temperature these changes are 23.9 % and 4.8 %. Both RCMs predicts a decrease of the standard deviation in precipitation and an increase of the standard deviation of temperature in the future (see Fig. 10c and 10d respectively). However the expected values of changes are significantly different. Both RCMs also predict significantly different changes in the skew coefficient of series (Fig. 10e and 10f). With respect the hydrology analysis, both RCMs predict significant decreases of mean streamflow (-43.5 % and -57.2 % for RCM2 and RCM9 respectively) (Fig. 11a). In the case of the standard deviation, the RCMs predict a reduction (Fig. 11b). The average change in monthly standard deviation is -26.2 % and -57.5 % for RCM2 and RCM9 respectively. In the case of the skew coefficient both RCMs show an increment with respect the historical scenario (Fig. 11c). We also analysed the coefficient of variation (ratio of the standard deviation to the mean) of historical and future series of precipitation, temperature, and streamflow (Table 4). Both RCMs predict an increase of the precipitation and streamflow variability, and a reduction of temperature variability. This increment in precipitation variability is also described in other climate change impact studies (Pendergrass et al., 2017; Polade et al., 2017).

Table 4: Coeficient of variation of the historical and future series generated from RCM2 and RCM9 for the precipitation, temperature, and streamflow.

|  | Coefficient of variation (CV) | | |
|---|---|---|---|
|  | Precipitation | Temperature | Streamflow |
| Historical | 0.80 | 0.46 | 0.69 |
| RCM2 | 1.07 | 0.41 | 0.84 |
| RCM9 | 1.10 | 0.42 | 1.07 |

*Line 211: "predict significant decreases of streamflow (-43.5 and 57.2%)" Should it be -57.2%?*

Thank you. We corrected it:

With respect the hydrology analysis, both RCMs predict significant decreases of mean streamflow (-43.5 % and -57.2 % for RCM2 and RCM9 respectively) (Fig. 11a).

*Lines 215-217: "In the case of the meteorological droughts the first SPI threshold for which droughts periods are detected in the historical scenario is -3.0. In the future scenarios this value is -5.2 and -4.6 for the RCM2 and RCM9 respectively". I think that it will be interesting to assess the changes in the parameters of the future distribution with regard to the historical one (even if only the historical distribution is used to obtain the future SPI).*

Done:

Significant changes are also expected for droughts. In the case of the meteorological droughts the first SPI threshold for which droughts periods are detected in the historical scenario is -3.0. In the future scenarios this value is -5.2 and -4.6 for the RCM2 and RCM9 respectively (Fig. 12). In order to perform an appropriate analyses of the future droughts with respect to the historical, the future SPI calculation were estimated by using the parameters of the gamma distribution obtained in the historical period (Collados-Lara et al., 2018). If the parameters of the gamma distribution were adjusted to the future series of values, the changes in the parameters would be significant. For RCM2 we would obtained $\alpha = 19.9$ and $\beta = 2.6$ (instead of the historical values $\alpha = 16.1$ and $\beta = 3.2$) and for RCM9 $\alpha = 19.0$ and $\beta = 2.7$ (instead of the historical values $\alpha = 16.1$ and $\beta = 3.2$).

*Lines 219-224: Check the signs of the SPI values.*

Done. When we refer to thresholds of SPI we used sing (-) and for statistics (intensity or magnitude) we used (+).

*Lines 231-257: in my opinion, the Discussion section does not address properly the limitations of the selected methodology (see my previous comments).*

We included a new section in Discussion to point the limitations and future works related to the proposed approach:

5.1 Hypothesis assumed, limitations and future works

Although we have demonstrated the utility of the proposed approach to assess future impacts on meteorological and hydrological droughts, we want to highlight some hypothesis and limitations assumed and to identify potential future research aligned with this study:

- We have used a bias correction method based on the assumption of bias stationarity of climate model outputs. However, this assumption may not be valid to study some problems due to the significance of the influence of climate variability on them. Other approaches should be explored to take into account non-stationarity bias of RCMs simulations (e.g. Hui et al., 2020).
- We have applied the same bias correction procedure for all the range values in accordance with the climatic variable distribution function. We did not consider

the impact of bias correction techniques on the tails of the distribution, which could be important to analyse extremes (Volosciuk et al., 2017).

- In this work a univariate bias correction method is used. It does not consider the dependence between precipitation and temperature which could be explored in future assessments. Meyer et al. (2019) found that incorporating or ignoring inter-variable relationships between temperature and precipitation could impact the conclusions drawn in hydrological climate change impact studies in alpine catchments.

- The streamflow information available for this case study cannot be divided into two long-enough (e.g. 30 years) series representative of the climate/hydrology to perform explicitly a validation of the bias correction models (Chen et al., 2021). We assumed that the statistics of any long-enough periods remain invariant. In this case the calibration implicitly could be considered validated, due to the fact that the same results would be obtained under this hypothesis for any other period representatives of the climate/hydrology conditions.

- In our case study the influence of temperature was considered only in the hydrological assessment by using rainfall-runoff models. However other meteorological droughts indices that consider temperature could be included in the analysis [e.g. the Standardised Precipitation-Evapotranspiration Index (SPEI) (García-Valdecasas Ojeda et al., 2021)].

- The corrected control simulation series obtained by using a quantile mapping bias correction presents a very good performance with respect the historical series in terms of basic statistics. In the case of droughts (calculated from SPI/SSI) the bias correction approach clearly improves the fit of the RCM simulation series to the historical series, but the performance is lower than for basic statistics. Other bias correction procedures should be explored to improve the performance for droughts statistics.

- The proposed method has not been tested in other typologies of basin, as for example in Alpine basins where snowmelt component may have a significant influence on the results.

---

## Author Comment (AC3)

July 16th, 2021

Dear Reviewer,

We are sending you the response to the referee comments (**nhess-2021-121-RC3**) of the manuscript -ref: **NHESS-2021-121-** and entitled "Do climate models that better approximate local meteorology improve the assessment of hydrological responses? Analyses of basic and drought statistics" by Antonio-Juan Collados-Lara, Juan-de-Dios Gómez-Gómez, David Pulido-Velazquez, and Eulogio Pardo-Igúzquiza.

We would like to express our sincere gratitude for your in-depth revision that will unquestionably help us to improve the manuscript. We have taken into account all the comments and we have provided response to them.

Thank you very much for your time and consideration.

Yours sincerely,

Antonio-Juan Collados-Lara, Juan-de-Dios Gómez-Gómez, David Pulido-Velazquez, and Eulogio Pardo-Igúzquiza.
Authors

*The manuscript evaluates the ability of 9 CORDEX RCMs to simulate meteorological (i.e., precipitation) and hydrological (i.e., streamflow) variables, as well as drought statistics, in the Cenajo basin (Southern Spain). The best RCMS are then used to generate future scenarios.*

*General comments*

*The manuscript is interesting and within the scope of the journal. From a methodological viewpoint, some relevant details need to be specified to understand the validity of the proposed approach, with special reference to drought analysis. Some sections should be re-organized. Also, the language must be improved in some parts.*

We thank the Reviewer for recognizing the interest and suitability of the paper for Natural Hazards and Earth System Science.

*Major comments*

*The title of the manuscript is wordy and redundant. Please rephrase.*

We modified the title:

Do climate models that better approximate local meteorology improve the assessment of hydrological responses? Analyses of basic and drought statistics

*LL 56-57: The authors state that "In literature few works analyze the reliability of RCMs considering meteorological droughts." Please add references to previous studies on this topic and highlight the main differences with your study. In particular, the manuscript would benefit from a comparison with a recent study by Peres et al. (https://doi.org/10.5194/nhess-20-3057-2020), dealing with a statistical methodological framework to assess the skill of the EURO-CORDEX RCMs to simulate historic climate (temperature and precipitation) and drought characteristics (duration, accumulated deficit, intensity, and return period), at seasonal and annual timescales, in Southern Italy.*

The main novelty is that in this work we also analyse the propagation of meteorological droughts to hydrological droughts. We have clarified it within the new version of the manuscript:

In literature few works analyse the reliability of RCMs considering meteorological droughts (Peres et al., 2020; Aryal and Zhu, 2021). In this work we also analyse the propagation of meteorological droughts to hydrological droughts. As far as we know, there are not studies that analyse if climate models that provide the best approximations of the local historical meteorology provide also better assessments of the hydrological impacts.

Added references:

Aryal, Y., Zhu, J.: Evaluating the performance of regional climate models to simulate the US drought and its connection with El Nino Southern Oscillation, Theor Appl Climatol, https://doi.org/10.1007/s00704-021-03704-y, 2021.

Peres, D. J., Senatore, A., Nanni, P., Cancelliere, A., Mendicino, G., and Bonaccorso, B.: Evaluation of EURO-CORDEX (Coordinated Regional Climate Downscaling Experiment for the Euro-Mediterranean area) historical simulations by high-quality observational datasets in southern Italy: Insights on drought assessment, Nat. Hazards Earth Syst. Sci., https://doi.org/10.5194/nhess-20-3057-2020, 2020.

*L 148: The authors have performed a lumped analysis in the Cenajo basin. To this end, they have to specify:*

*If the reference grids of both the historical data and the CORDEX simulations are equivalent; If not, how do they pair the information from the two grids?*

*how many grid cells fall within the Cenajo basin;*

*If the gridded historical and simulated precipitation data are spatially aggregated at the basin scale level and how.*

We clarified these points in the new version:

We used historical climatic data (precipitation and temperature) provided by Spain02 v2 dataset (Herrera et al., 2012) for the period 1972-2001. In this work we performed a lumped analysis in the Cenajo basin. The RCMs were retrieved from the CORDEX project (CORDEX PROJECT, 2013), with a spatial resolution of 0.11º (approximately 12.5 km). Note that Spain02 dataset uses the same reference grids than CORDEX project. The most pessimistic emission scenario (RCP8.5) for the future horizon 2071-2100 was selected for the future projections. For this scenario we analysed nine RCMs corresponding to four different General Circulation Models (GCMs) (see Table1). In our case study 33 cells of the grid mesh fall within the basin. The historical and simulated (from RCMs) precipitation and temperature were aggregated at basin scale considering a weighted average value according to the area of each grid mesh inside the basin. We also used official monthly natural streamflow data within the Cenajo basin for the historical period 1972-2001 (adopted as reference). These data were taken from the available information coming from the Spanish Ministry for the Ecological Transition and the Demographic Challenge.

*The authors apply the SPI for meteorological drought analysis. However, it is not clear which time scale is used to aggregate monthly precipitation (1, 2, 3 months?) and which probability distribution is fitted to such data (gamma distribution?) for SPI computation. The authors should be aware that if they simply calculate the standard normal values corresponding to the differences of monthly precipitation data and the related monthly means, divided by the related monthly standard deviations, they do not obtain SPI, but another index known as the Standardized Rainfall Anomaly (Jones and Hulme, 1996), which is equal to the SPI only if aggregated precipitation data are normally distributed.*

We calculated SPI using an aggregation time of 12 months and the calculation method is comprised of a transformation of one frequency distribution (gamma) to another standardized frequency distribution (normal). The same procedure was applied to streamflow to obtain the SSI. We have clarified it in the new version:

The meteorological and hydrological drought analysis was developed by applying the Standard Precipitation index (SPI) (Bonaccorso et al., 2003; Livada and Assimakopoulos, 2007) and Standard Streamflow index (SSI) (Salimi et al., 2021), respectively. They were estimated for periods of aggregation equal to 12 months. The calculation method requires the transformation of a gamma frequency distribution function to a normal standardized frequency distribution function. The statistics of the SPI/SSI series are obtained by applying the run theory (González and Valdés, 2006; Mishra et al., 2009) for different SPI/SSI thresholds from the lower SPI/SSI to 0.

*Moreover, once that the SPI series is computed by using different threshold values, drought characteristics such as frequency, length, magnitude, and intensity are determined. The authors do not clarify how these characteristics are computed. Nonetheless, I believe that, for instance, drought magnitude for drought events longer than one month has been computed as the sum of SPI values over the length. Is it correct? If so, the approach is misleading since a SPI value already quantifies the magnitude of a dry or a wet period occurs during the considered aggregation period.*

We calculated the magnitude for each drought event as the sum of the SPI values over the length of this event but the magnitude associated to a given threshold was calculated as the average of the magnitude for all the droughts events for this thresholds. We have clarified it and the calculation of the others droughts statistics in the new version:

The frequency is defined as the number of droughts events for each SPI threshold. For each drought event, we assess its duration as the number of months that the SPI is below a given threshold, its magnitude as the summation of the SPI values for each month of the event, and its intensity as the minimum SPI value. For each threshold we estimate the mean duration, magnitude, and intensity as the mean values of the cited variables for all the drought events.

*The number of drought events identified for each considered threshold should be indicated in a table, together with the mean values of the corresponding characteristics. I am afraid that for the control scenario, very few droughts are identified for threshold values corresponding to severe and extremely dry conditions. Thus, I wonder how fair could be the comparison between observations and simulations? In addition, if the analysis is lumped (i.e., a single series for the whole basin is considered for each variable), it would be interesting to ascertain whether the drought statistics evaluated on RCM simulations correspond to the same drought events identified on the historical series.*

The number of drought (frequency) is also represented in the figures together with mean length, mean magnitude, and mean intensity. The number of droughts of the control scenario (Fig.8a) is different depending on the RCM, but, after the correction (some of them are similar to the historical), all simulations have similar values of number of droughts (Fig. 8b).

On the other hand, the droughts events identified for the historical series do not correspond to the same events in the corrected control series. Note that RCM simulations do not simulate specific days or months. They simulate climate, providing plausible meteorological series for a specific climate conditions. The bias correction

approach is also designed to obtain the climate change signal (the statistics of the series) but not to reproduce the historical series from the control simulations. For example we represented in Figure RC3-1 the months in the historical period with SPI values below zero, for the historical and corrected control simulation series, and the obtained events are different.

[Figure]

Figure 8: Drought statistics (frequency, length, magnitude and intensity) of the period (1972-2001) for the historical and control simulation series (left column) and historical and corrected control simulation series (right column) for precipitation (meteorological droughts).

[Figure]

Figure RC3-1. Months in which the SPI value is below zero for the historical and corrected control simulations series for the used RCMs.

*Finally, bias correction through quantile mapping applied to SPI (if precipitation) or to SSI (if streamflow) series is a little confusing since these series are standard normal distributed by definition, therefore I do not expect big differences between the historical series and the control simulation series, unless due to sampling variability. Please clarify this point and explain the results illustrated in Figures 7 and 8.*

The corrected control series generated by applying a quantile mapping bias correction to the RCMS simulations show a very good fit with respect the historical series in terms of basic statistics. In terms of droughts (calculated from SPI), the bias correction approach clearly improves the fit of the RCM simulation series) to the historical series but the performance is lower than the one obtained for basic statistics. The left panel of the Figures represents the statistics before bias correction and the right panel after bias correction .The objective of this work is to classify RCMs according its capacity to reproduce historical basics and droughts statistic. Other bias correction technique should be explored to improve the performance for droughts statistics. We have introduced some changes in the new version of the manuscript in order to clarify it.

In the case of meteorological droughts (calculated from SPI) the bias correction approach clearly improves the fit of the RCM simulation series to the historical series for the four considered statistics (frequency, duration, magnitude and intensity). Note the differences between the left panel of Fig. 8 (control simulation and historical series) and right panel of Fig. 8 (corrected control simulation and historical series). For frequency the mean of SE for all the RCMs before the correction is 0.69 and after the correction is 0.23. For duration, magnitude and intensity these values are respectively 0.51 vs. 0.17, 0.88 vs. 0.30 and 0.38 vs. 0.13. In the same way, hydrological droughts were studied considering the SSI. Significant improvements are also observed for hydrological droughts (Fig. 9) after the bias correction procedure: frequency (mean SE of 0.63 vs. 0.34), duration (mean SE of 0.50 vs. 0.23), magnitude (mean SE of 0.83 vs. 0.51), and intensity (mean SE of 0.48 vs. 0.15). The left panel represents the droughts statistics of the historical and control series before applying the bias correction technique and the right one after a bias correction approach.

We also stated in the new section "5.1 Hypothesis assumed, limitations and future works" the necessity of exploring additional bias correction techniques to improve the performance for droughts statistics:

The corrected control simulation series obtained by using a quantile mapping bias correction presents a very good performance with respect the historical series in terms of basic statistics. In the case of droughts (calculated from SPI/SSI) the bias correction approach clearly improves the fit of the RCM simulation series to the historical series, but the performance is lower than for basic statistics. Other bias correction procedures should be explored to improve the performance for droughts statistics.

*Minor comments*

*L 10: Hydrological impacts of what? Maybe, change with "hydrological response".*

Done:

This work studies the benefit of using more reliable local climate scenarios to analyse hydrological responses.

*LL 10-12: "It assumes that … when they provide better approximation to the historical basic and drought statistics." This sentence is rather unclear and must be rephrased.*

Done:

It assumes that Regional Climate Models (RCMs) simulations are more reliable when they provide better approximations to the historical basic and drought statistics after applying bias correction to them.

*LL 18-20: In the last sentence there is no reference to the future scenarios of hydrological droughts.*

Done:

These two RCMs also predict higher changes in mean streamflow (-43.5 and -57.2 %) and hydrological droughts. The two RCMs also predict worrying changes in streamflow (-43.5 % and -57.2 %) and hydrological extreme droughts: frequency (from 3 to 11 and 8 events), length (8.3 to 15.4 and 29.6 months), magnitude (from 3.98 to 11.84 and 31.72 SSI), and intensity (0.63 to 0.90 and 1.52 SSI).

*LL 103 and 114: The term "goodness of fit" is usually applied to describe how well a statistical model (e.g., a probability distribution) fits a set of observations. I am not sure it is appropriate for RCM simulations.*

We changed this term by performance along the manuscript.

*L 162 and L 192: Sections 4.2 and 4.3 have the same title. Merge the two sections.*

It was an error. We changed the title of Section 4.3 to "4.3 Classification of RCMs"

*L 219: "the threshold of "-" 1.7 of SPI (considered to define extreme droughts …)". Usually, -2 is used for extreme droughts.*

We used the -1.7 SPI to identify extreme droughts because the Droughts Plan of the Segura River basin authority (where the Cenajo basin is located) proposes to use this value, but, in the figures we also show results in the different figures for all the thresholds.

---

## Referee Report (RR1)

**Comments to "Do climate models that are better at approximating local meteorology also improve the assessment of hydrological responses? An analysis of basic and drought statistics"**

**1  Overview**

The authors study whether RCM simulations that provide the best approximations of the local meteorology also provide the best assessments of the local hydrological impact. The authors propose a methodology that briefly follows these steps: the bias in RCM control simulations is corrected, hydrological series are estimated through a rainfall-runoff model those inputs are the bias-corrected RCM meteorological data, the RCM models are classified, and lastly, local future climate scenarios are generated with the best RCM models and results are analyzed.

The originality of the paper, as well as the scientific soundness, is suitable for Nat. Hazards Earth Syst. Sci. The paper is well-written and well-organized. I recommend the publication after some minor changes.

**2  Observations**

My main observations are:

- Eq. (1). The equation should be written as follows:

$$SE = \frac{1}{\left(\frac{1}{N}\sum_{i=1}^{N}S_{h,i}\right)^2}\frac{1}{N}\sum_{i=1}^{N}\left(S_{c,i}-S_{h,i}\right)^2,\tag{1}$$

- The Case study is the Cenajo basin. The authors should indicate whether there are dams in the studied area since dams must be taken into account in the rainfall-runoff model. If there are any dams, the authors should explain how they are taken into account.

- Line 191. Is the period 1972-2001, or 2071-2100?

- Line 301. The authors state that they have **demonstrated** in a case study that the corrected... I do not totally agree with them on this sentence. They don't demonstrate that their methodology will provide the best results as they do not check their approach with many basins. I rather write the statement as they have **shown** in a case study... This comment also applies to line 348.

**3 Typos**

There are some typos in the paper. Some are:

- Line 42. Although in  recent years.

- Line 43. please, add a space between "increased" and the parenthesis.

- Line 136. Please, remove the s at the end of "respect".

- Line 147. Please, change "to" for "with", i.e. "in accordance  with this total".

- Line 233. Please, add a hyphen between "best" and "corrected".

- Line 243. Please, rewrite "the impact of climate variables  on streamflow".

- Line 245. Please, rewrite "an increase  in"

- Line 248. Please, add a space between "in" and "the".

- Line 251 Please, rewrite "decreases  in".

- Line 284. Please, add an article to the sentence: "It is accepted in **the** scientific community ...".

- Line 297. Please, add an space between "up" and "in".

- Line 322. Please, rewrite "could be important  to analyse".

- Line 340. Please, rewrite "the performance  of".

- Reference. Please, revise the list of references. The journal names of some references are missed. For instance, references in lines 383, 393, 403, 405, 436, 437, 441, 470, 484, and 491. moreover, doi link in the reference of line 405 does not work.

---

## Author Response (AR2)

21 Jan 2022

Executive Editor decision: Publish subject to minor revisions (review by editor)
by Joaquim G. Pinto

Comments to the author:

Dear authors, thanks for the revised version of the paper.

Both reviewers state that the paper has improved, but they still have some comments
that need to be taken care of before the manuscript can be accepted for publication in
NHESS. In particular, I agree that the title is indeed too long and should be revised.

Please revise the manuscript accordingly and provide the corresponded replies.

best regards,
Joaquim Pinto
(liason editor)

January 26th, 2021

Dear Editor and Reviewers,

We are sending you the second revised version of the manuscript -ref: **NHESS-2021-121-** and entitled (new title) "An approach to identify the best climate models for the assessment of climate change impacts on meteorological and hydrological droughts" by Antonio-Juan Collados-Lara, Juan-de-Dios Gómez-Gómez, David Pulido-Velazquez, and Eulogio Pardo-Igúzquiza.

We would like to express our sincere gratitude for your in-depth revision that unquestionably helps us to improve the manuscript as well as the opportunity given by the Editor to submit a revised version.

We have taken into account all the comments raised by the reviewers and we have provided explanations with our answers to the reviewers' comments in the response document. The new version of the manuscript was also reviewed by a professional English translator.

Thank you very much for your time and consideration.

Yours sincerely,

Antonio-Juan Collados-Lara
Corresponding author
Geological Survey of Spain (IGME)
Ríos Rosas 23
28003 Madrid
Spain
E-mail address: ajcollados@gmail.com

**REVIEWER #3 (Report #1)**

*Dear authors,*

*Thanks for your reply and efforts to improve your work also based on my previous comments.*

*My only criticism is that the title is still too long and discordant. Besides, "basic statistics" sounds too vague. Please make a further little effort to improve the title.*

We thank the Reviewer for recognizing efforts to improve your work and for the recommendation of publication. We modified the title according the suggestion of the Reviewer:

An approach to identify the best climate models for the assessment of climate change impacts on meteorological and hydrological droughts

**REVIEWER #4 (Report 2)**

*1 Overview*

*The authors study whether RCM simulations that provide the best approximations of the local meteorology also provide the best assessments of the local hydrological impact. The authors propose a methodology that briefly follows these steps: the bias in RCM control simulations is corrected, hydrological series are estimated through a rainfall-runoff model those inputs are the bias-corrected RCM meteorological data, the RCM models are classified, and lastly, local future climate scenarios are generated with the best RCM models and results are analyzed. The originality of the paper, as well as the scientific soundness, is suitable for Nat. Hazards Earth Syst. Sci. The paper is well-written and well-organized. I recommend the publication after some minor changes.*

We thank the Reviewer for recognizing the quality of the paper and for the recommendation of publication. We have taken into account all the comments raised by the Reviewer below.

*2 Observations*

*My main observations are:*

*• Eq. (1). The equation should be written as follows:*

$$ SE = \frac{1}{\left(\frac{1}{N} \sum_{i=1}^{N} S_{h,i}\right)^2} \frac{1}{N} \sum_{i=1}^{N} \left(S_{c,i} - S_{h,i}\right)^2 $$

Done.

*• The Case study is the Cenajo basin. The authors should indicate whether there are dams in the studied area since dams must be taken into account in the rainfall-runoff model. If there are any dams, the authors should explain how they are taken into account.*

Thanks to the reviewer comment we have realized that we did not explain well this point. The rainfall-runoff model provides runoff series in natural regime. We have used historical runoff data in natural regime by using the streamflow series from the SIMPA model to calibrate the Témez model. SIMPA is the model used by the water authorities in Spain for water planning. It was calibrated previously by restoring the gauge stations to the natural regime. So we have assessed the impacts on the streamflow series in natural regime (available resources) in the basin, thus not having to take into account the dams in the basin. We have corrected the text in section 3:

We also used official monthly natural streamflow data within the Cenajo basin for the historical period 1972-2001 (adopted as reference). The SIMPA model streamflow series (Alvarez et al., 2005) were used as historical data for calibration, due to the highly altered flow regime measured in gauge stations within this basin. Note that in the studied basin there are several dams. SIMPA is the model used by the water authorities in Spain for water planning. It was calibrated previously by restoring the gauge stations to the natural regime. Therefore, we assessed inflow scenarios in natural flow regime in the basin. These data were taken from the available information from the Spanish Ministry for Agrarian Development and Irrigation.

And added reference:

Alvarez J., Sanchez A., Quintas L. (2005). SIMPA, a GRASS based tool for Hydrological Studies. Proceedings of the FOSS/GRASS Users Conference - Bangkok, Thailand, 12-14 September 2004. International Journal of Geoinformatics. Volume 1, no 1 march 2005. Association for Geoinformation Technology.

• *Line 191. Is the period 1972-2001, or 2071-2100?*

Corrected:

The model was used to propagate the impact of climate variables on the streamflow between 2071 and 2100, a 30-year horizon, which is a period of time usually used in climate change analysis.

• *Line 301. The authors state that they have demonstrated in a case study that the corrected... I do not totally agree with them on this sentence. They don't demonstrate that their methodology will provide the best results as they do not check their approach with many basins. I rather write the statement as they have shown in a case study...*

Done:

We have shown in a case study that the corrected RCM simulations that provide the best approximations of the meteorological statistics also provide the best approximations for the hydrology.

*This comment also applies to line 348.*

Done:

We have also shown that the corrected RCM simulations that provide the best approximations of the meteorology also provide the best assessments of the hydrological impact.

*3 Typos*

Thank you. All the typos were corrected in the new version of the paper

*There are some typos in the paper. Some are:*

*• Line 42. Although in the recent years.*

*• Line 43. please, add a space between "increased" and the parenthesis.*

*• Line 136. Please, remove the s at the end of "respect".*

*• Line 147. Please, change "to" for "with", i.e. "in accordance to with this total".*

*• Line 233. Please, add a hyphen between "best" and "corrected".*

*• Line 243. Please, rewrite "the impact of climate variables to on streamflow".*

*• Line 245. Please, rewrite "an increase of in"*

*• Line 248. Please, add a space between "in" and "the".*

*• Line 251 Please, rewrite "decreases of in".*

*• Line 284. Please, add an article to the sentence: "It is accepted in the scientific community ...".*

*• Line 297. Please, add an space between "up" and "in".*

*• Line 322. Please, rewrite "could be important for analysing to analyse".*

*• Line 340. Please, rewrite "the performance for of".*

*• Reference. Please, revise the list of references. The journal names of some references are missed. For instance, references in lines 383, 393, 403, 405, 436, 437, 441, 470, 484, and 491. moreover, doi link in the reference of line 405 does not work.*